# WIMFRIS: WIndow Mamba Fusion and Parameter Efficient Tuning for Referring Image Segmentation

**Seunghun Moon**[*1], **Hyunwoo Yu**[*1], **Haeuk Lee**[*2], **Suk-Ju Kang**[†1]
[1]Department of Electronic Engineering, [2]Department of Artificial Intelligence
[1,2]Sogang University, South Korea
{moonsh97,hyunwoo137,leemail,sjkang}@sogang.ac.kr

## ABSTRACT

Existing Parameter-Efficient Tuning (PET) methods for Referring Image Segmentation (RIS) primarily focus on layer-wise feature alignment, often neglecting the crucial role of a neck module for the intermediate fusion of aggregated multi-scale features, which creates a significant performance bottleneck. To address this limitation, we introduce WIMFRIS, a novel framework that establishes a powerful neck architecture alongside a simple yet effective PET strategy. At its core is our proposed Hierarchical Mamba Fusion (HMF) block, which first aggregates multi-scale features and then employs a novel Window Mamba Fuser (WMF) module to perform effective intermediate fusion. This WMF module leverages non-overlapping window partitioning to mitigate the information decay problem inherent in State-Space Models (SSMs) while ensuring rich local-global context interaction. Furthermore, our PET strategy enhances primary alignment with a Mamba Text Adapter (MTA) for robust textual priors, a Multi-Scale Aligner (MSA) for precise vision-language fusion, and learnable emphasis parameters for adaptive stage-wise feature weighting. Extensive experiments demonstrate that WIMFRIS achieves new state-of-the-art performance across all public RIS benchmarks. Code is available here.

## 1 INTRODUCTION

Referring Image Segmentation (RIS) is a dense prediction task that generates precise pixel-level masks for objects described in natural language. Unlike semantic segmentation Xie et al. (2021); Yan et al. (2024); Fu et al. (2025); Shim et al. (2023); Kang et al. (2024); Yu et al. (2024) which classifies each pixel into a fixed set of categories, RIS requires jointly modeling capacity of fine-grained visual content and open-vocabulary text semantics to localize the target instance. While early RIS methods Li et al. (2018); Liu et al. (2017) utilize separate backbones with late fusion mechanisms like fully connected or recurrent modules, frameworks following transformer architectures Dosovitskiy et al. (2020) leverage the cross-attention mechanism for more precise and robust vision–language alignment.

Another line of work Xu et al. (2023); Wang et al. (2024b); Huang et al. (2025) has achieved great success by employing Parameter-Efficient Tuning (PET) approaches to adapt large, pre-trained backbones such as CLIP Radford et al. (2021) or DINO Caron et al. (2021); Oquab et al. (2023) for the RIS task. These methods utilize lightweight trainable modules to efficiently adapt the powerful, pre-trained features, while keeping the vast majority of the backbone parameters frozen. These modules are specifically designed to efficiently refine and align the backbone's rich visual representations with the fine-grained linguistic cues of the referring expression, enabling precise pixel-level localization.

Despite their effectiveness, existing PET-based approaches predominantly focus on designing specific adapters to align textual and visual features from corresponding encoder layers. However,

---

[*]Equal contribution.
[†]Corresponding author.

Table 1: Analysis of neck module functioning. Omission of the neck module results in significant performance degradation, highlighting its critical role in performing intermediate fusion. Incorporating our HMF-block into existing PET-based approaches further improves their performance. Furthermore, our WIMFRIS utilizing proposed HMF-block and PET framework achieves state-of-the-art performance.

| Text Enc. | Vision Enc. | PET Method | # Learnable PET Params | Neck | RefCOCO | | |
|---|---|---|---|---|---|---|---|
| | | | | | val | testA | testB |
| CLIP-ViT-B/16 | CLIP-ViT-B/16 | ETRIS | 1.4M | ✗ | 72.2 | 73.9 | 70.8 |
| | | | | ETRIS | 74.5 | 76.5 | 72.9 |
| | | | | **Ours** | **75.7** | **77.5** | **73.3** |
| | DINOv2-B/14 | DETRIS | 1.4M | ✗ | 74.3 | 75.8 | 70.8 |
| | | | | DETRIS | 75.8 | 77.7 | 72.9 |
| | | | | **Ours** | **76.4** | **78.3** | **73.6** |
| | | **Ours** | 1.4M | **Ours** | **77.2** | **78.9** | **74.3** |

these adapters are designed for basic, layer-wise fine-tuning, providing only a generic primary alignment. The task-specific intermediate fusion, which requires aggregating and contextualizing these individually aligned features from across all layers, is therefore a more distinct and critical process. This highlights the importance of the neck architecture performing intermediate fusion. Consequently, the absence of a carefully designed neck can cause an information bottleneck, preventing the full potential of the rich, multimodal features from being fully exploited and ultimately hindering performance. As shown in Table 1, the omission of a neck module leads to a substantial performance degradation, demonstrating the importance of not only performing layer-wise tuning, but also fusing the resulting features to achieve fine-grained, task-specific feature.

This paper proposes a novel neck architecture, namely Hierarchical Mamba Fusion (HMF) block, that enables powerful intermediate vision-language modality fusion. Our module can be easily integrated into existing PET-based RIS methods and enhances the performance of existing approaches, as shown in Table 1. Existing neck architectures are suboptimal because they are designed to redundantly repeat the basic modality alignment already handled by the PET adapters at each layer. In contrast, as illustrated in Fig. 1, rather than fusing layer-wise features separately, our HMF block first aggregates the individually-tuned visual features from backbone layers into a unified, multi-semantic representation. It then performs a single, powerful intermediate fusion

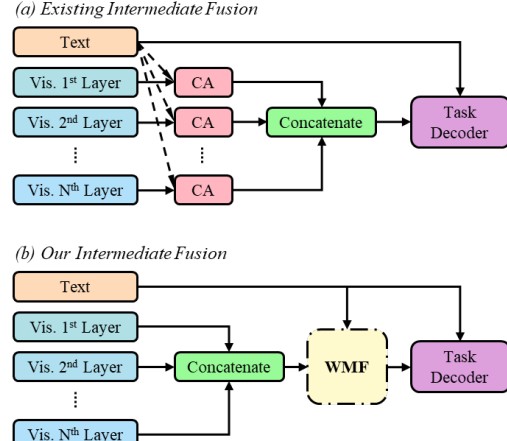

Figure 1: Comparison between previous intermediate fusion and our approach. Instead of fusing layer-wise visual features with text prior separately using conventional cross-attentions (a), we first aggregate them for unified, multi-semantic visual representation and conduct modality fusion via our proposed WMF module in HMF neck (b).

with linguistic cues with our novel Mamba-based Gu & Dao (2023) Window Mamba Fuser (WMF) module. Specifically, WMF module partitions the visual feature map into non-overlapping windows and prepends a shared global textual prior to each window sequence. WMF processes these compact windowed subregions in parallel with a Mamba layer, effectively constraining the overall sequence length. This mitigates the exponential decaying problem Wang et al. (2024a), where SSMs inherently bias toward recent tokens as repeated hidden state updates attenuate information from distant sequential positions exponentially, thereby enabling effective modality fusion. Additionally, attaching the global textual token to every window sequence ensures that each local region directly interacts with global linguistic context.

Moreover, we introduce a simple yet effective PET framework designed to empower the backbone adapters to more effectively perform their role of primary alignment. Specifically, we design the Mamba Text Adapter (MTA), text adapter utilizing the State-Space Model (SSM)-based Mamba block that provides an enhanced global textual prior with full sequence awareness and linear computational complexity. Furthermore, we design the Multi-Scale Aligner (MSA), a vision adapter that leverages strip-convolutional Receptive Field Mixer (RFMixer) module to extract features across multiple receptive fields. It also employs a multi-head cross-attention layer to interact with the

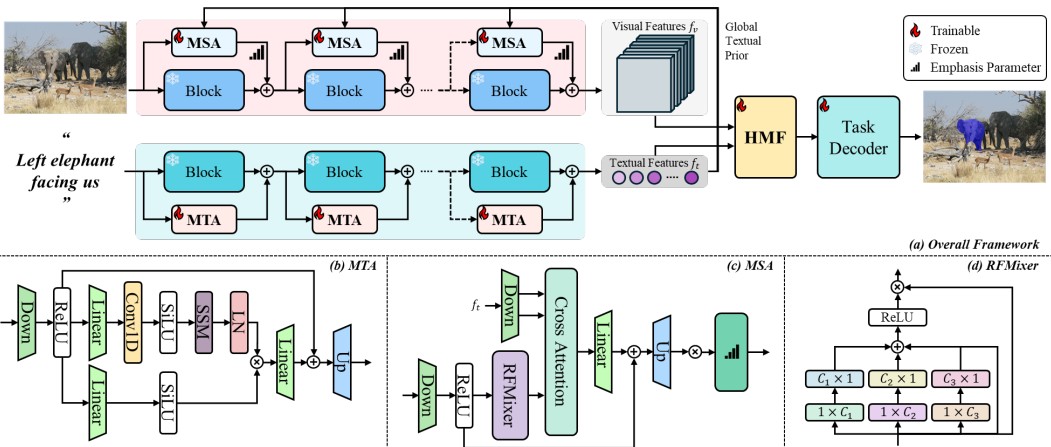

Figure 2: (a) Overview of WIMFRIS architecture. Frozen CLIP text encoder layers and DINOv2 vision encoder layers are parameter-efficient tuned by MTA (b) to get enhanced global textual features $f_t$, and MSA (c) with learnable emphasis parameters and RFMixer (d) to obtain fine-grained visual features $f_v$. Subsequently, our HMF block performs powerful vision-language intermediate modality fusion.

enhanced global textual prior, thereby improving primary modality fusion. Lastly, we adopt a learnable emphasis parameter before each vision adapter to consider the different contributions of each backbone stage. This allows the model to assign an independent and adaptive contribution to each stage during training, ensuring that more informative stages can exert a greater impact on the fused features. With only 1.3% to 2.2% backbone parameter updates, the proposed framework achieves a more robust global textual prior and enables more effective layer-wise modality fusion.

Incorporating the proposed HMF block and the PET strategy, we introduce a novel framework for RIS, namely **WIMFRIS** (**WI**ndow **M**amba **F**usion and parameter-efficient tuning for **RIS**). Our main contributions can be summarized as follows:

- We find that existing PET-based RIS approaches focus on layer-wise modality fusion and neglect the importance of aggregated intermediate modality fusion, and we propose a novel HMF block designed for effective intermediate modality fusion with its core WMF module.

- We propose a PET strategy that combines an SSM-based MTA for strong global textual priors, an MSA with a RFMixer module and cross-attention layer for precise vision–language fusion, and learnable emphasis parameters to adaptively weight each backbone stage.

- We introduce WIMFRIS, a novel RIS framework leveraging our HMF and PET strategy that achieves new state-of-the-art performance in every RIS benchmarks.

## 2 METHODOLOGY

### 2.1 ARCHITECTURE OVERVIEW

As shown in Fig. 2 (a), our proposed WIMFRIS is a referring image segmentation framework leveraging simple yet effective Parameter-Efficient Tuning (PET) strategy and powerful intermediate fusion neck block. The core of WIMFRIS is the novel Hierarchical Mamba Fusion (HMF) block visualized in Fig. 3 (b), an SSM-based intermediate vision-language modality fuser that aligns global textual priors to window-partitioned local visual features with proposed Window Mamba Fuser (WMF) module. In addition, we freeze the pretrained CLIP Radford et al. (2021) text encoder and DINOv2 Oquab et al. (2023) encoder, then fine-tune them using two PET modules. The Mamba Text Adapter (MTA) employs an SSM-based Mamba block Gu & Dao (2023) to consolidate long-range dependencies across text tokens into a global textual prior. The Multi-Scale Adapter (MSA) aligns stage-wise visual features with that enhanced prior through the RFMixer module and cross attention layer while using learnable emphasis parameters to adaptively modulate its contribution.

## 2.2 Parameter-Efficient Tuning Strategy

Although DINO encoder is powerful, its unimodal pretraining lacks visual-text aligning capacity. So we employ a lightweight MSA adapter module that leverages outputs of the CLIP text encoder to parameter-efficient tune DINO vision encoder with effective modality alignment. However, directly using CLIP outputs without further refinement is suboptimal for RIS, which inherently requires precise global contextual text representations. Thus, we utilize a Mamba-based MTA module that explicitly strengthens the long-range dependencies among text tokens. By doing so, MTA generates an enhanced global textual prior, which is then fused with layer-wise features of DINO vision encoder via our carefully designed MSA.

**Mamba Text Adapter (MTA).** As illustrated in Fig. 2 (b), MTA produces text token which can be leveraged as a strong referring context guidance for modality alignment. Given a text token $\mathbf{x}_t^l \in \mathbb{R}^{L_t \times C_l}$ where $L_t$ and $C_l$ denote the sequence length and the channel dimension at the $l$-th layer of the CLIP text encoder respectively, the MTA produces a refined output $\mathbf{x}_{t\_out}^l$ with enhanced long-range contextual dependencies. Our MTA can be mathematically formulated as follows:

$$
\begin{aligned}
\mathbf{x}_{t\_fc}^l &= \sigma_{\text{ReLU}}(\text{Down}(\mathbf{x}_t^l)), \\
\mathbf{x}_{t\_SSM}^l &= \text{SSM}\big(\sigma_{\text{SiLU}}(\text{Conv1D}(\text{Proj}_{\text{in}}(\mathbf{x}_{t\_fc}^l)))\big), \\
\mathbf{x}_{t\_res}^l &= \sigma_{\text{SiLU}}(\text{Proj}_{\text{in}}(\mathbf{x}_{t\_fc}^l)), \\
\mathbf{x}_{t\_out}^l &= \text{Up}\big(\text{Proj}_{\text{out}}(\mathbf{x}_{t\_SSM}^l \cdot \mathbf{x}_{t\_res}^l) + \mathbf{x}_{t\_fc}^l\big),
\end{aligned}
\tag{1}
$$

where $\sigma_{ReLU}(\cdot)$ and $\sigma_{SiLU}(\cdot)$ denote the ReLU and SiLU activations, respectively. $\text{Down}(\cdot)$ and $\text{Up}(\cdot)$ are the down-sampling and up-sampling linear operations that reduce and restore the channel dimension of the input. $\text{Proj}_{\text{in}}(\cdot)$ and $\text{Proj}_{\text{out}}(\cdot)$ denote the input and output projection layers for the hidden state of the SSM. $\text{Conv1D}(\cdot)$ is a 1D convolution. By leveraging SSM-based scanning mechanism, MTA captures rich global text semantics with linear computational complexity, ensuring efficient modeling of long-range dependencies. In addition, our MTA consistently improves vision-language modality alignment by furnishing enhanced global textual prior to modality fusing modules.

**Multi-Scale Aligner (MSA).** The proposed MSA is designed, as illustrated in Fig. 2 (c) and (d), around two core components, the RFMixer and the cross-attention block. This adapter module takes as input the $i$-th layer vision feature $\mathbf{x}_v^i \in \mathbb{R}^{H \times W \times C_i}$, where $H$, $W$ and $C_i$ refer to the height, width and the corresponding layer channel dimensions, respectively. It also receives the textual features $f_t$ produced by the MTA-fine-tuned CLIP text encoder as global textual prior that enriches the vision-language modality alignment. The RFMixer consists of multiple branches of depth-wise strip convolutions, enabling it to efficiently capture multi-scale contextual information. Specifically, RFMixer can be defined as follows:

$$
\begin{aligned}
\tilde{\mathbf{x}}_v^{i(k)} &= \text{DWConv}_{c_k \times 1}\Big(\text{DWConv}_{1 \times c_k}(\mathbf{x}_v^i)\Big), \\
\tilde{\mathbf{x}}_v^i &= \sigma_{\text{ReLU}}\Big(\mathbf{x}_v^i + \sum_{k=1}^3 \tilde{\mathbf{x}}_v^{i(k)}\Big), \\
\mathbf{x}_{v\_mix}^i &= \tilde{\mathbf{x}}_v^i \odot \mathbf{x}_v^i,
\end{aligned}
\tag{2}
$$

where $\text{DWConv}_{c_1 \times c_2}(\cdot)$ denotes the depth-wise 2D convolution with kernel size $(c_1, c_2)$, and $\odot$ refers to the Hadamard product. By mixing depth-wise convolutions of varied kernel sizes, the RFMixer aggregates information across diverse receptive fields, capturing both fine-grained details and broader context in a single, efficient module. This multi-scale interaction strengthens the visual features, making them more effective for the subsequent alignment with linguistic cues. We then leverage the RFMixer-enriched visual feature $\mathbf{x}_{v\_mix}^i$ as the query in a cross-attention block, with the MTA-enhanced global textual prior $f_t$ serving as both key and value to obtain the aligned feature $x_{v\_fused}^i$. This mechanism enables more robust modality alignment and effectively integrates linguistic context into the visual representation.

**Emphasis Parameter.** As numerous studies Lin et al. (2017); Raghu et al. (2021); Ghiasi et al. (2022); Vilas et al. (2023) have demonstrated that multi-layer visual backbones capture distinct layer-wise contextual information. Therefore, simply summing each feature map and its adapter

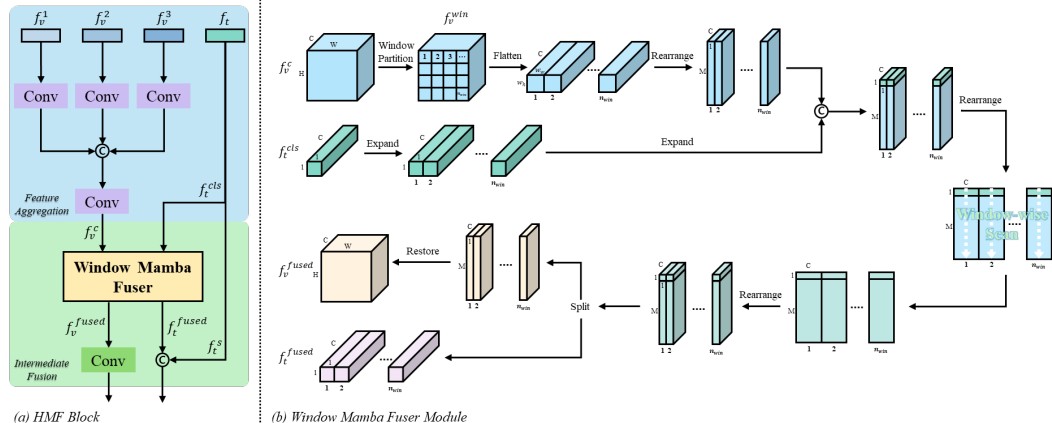

Figure 3: (a) Our HMF block designed for robust and effective intermediate vision-language modality fusion. It first aggregates the multi-layer vision features using convolutional layers and fuses them with global text class token with WMF module. (b) The WMF module partitions the vision–text feature map into non-overlapping windows and prepends a shared global text class token to each window. It then performs a window-wise SSM scans and reassembles the fused outputs through attentional gating.

output is suboptimal, as some layers require stronger fine-tuning than others. To address this, we adopt a simple learnable emphasis parameter that adaptively modulates the contribution of each layer when applying MSA during fine-tuning phase. We parameterize the emphasis gating via a scalar logit $p$ and a sigmoid activation, initialized as $\sigma(p) = \alpha_0$. The learnable emphasis parameter is mathematically defined as follows:

$$
\begin{aligned}
p &= \mathrm{logit}(\alpha_0) = \ln\left(\frac{\alpha_0}{1 - \alpha_0}\right), \\
\alpha &= \sigma_{Sigmoid}(p) = \frac{1}{1 + e^{-p}} \in (0, 1), \\
f_v^i &= x_v^i + \alpha \cdot \mathbf{x}_{v\_fused}^i,
\end{aligned}
\tag{3}
$$

where $f_v^i$ denotes the stage-$i$ fused visual feature. Rather than naively utilizing single trainable parameter, this gives us two key advantages. First, the sigmoid activation ensures the numerical stability that $\alpha$ always lies strictly between 0 and 1, preventing the adapter branch from exploding or inverting. Second, it enables adaptive modulation of the fine-tuning strength for each stage, ensuring that the stages capturing different contextual information can be fine-tuned dynamically as required.

## 2.3 HIERARCHICAL MAMBA FUSION (HMF) BLOCK

Our HMF block, as shown in Fig. 3 (a), is designed to conduct robust and effective intermediate vision-language modality alignment. HMF block first aggregates these features into a single fine-grained visual representation. It then fuses this representation with the textual class token via our carefully designed SSM-based Window Mamba Fuser (WMF) module. The WMF module partitions the visual features into non-overlapping windows, effectively constraining the spatial sequence length. After a global textual class token is prepended to each window, the resulting sequences pass through the Mamba layer for strong, window-wise local-global modality alignment.

**Feature Aggregation.** The feature aggregation step consolidates independently processed multiscale information into a single, unified feature map. Specifically, we first extract three feature maps, $f_v^i$, $i = \{1, 2, 3\}$, from two intermediate layers and the final output of the MSA-tuned DINOv2 Oquab et al. (2023) vision encoder. These features are then concatenated along the channel dimension and projected through a $1 \times 1$ convolution, yielding a unified feature map $f_v^c \in \mathbb{R}^{H \times W \times C}$. The rich multi-scale contextual information consolidated in this feature map allows the WMF module to perform a more robust modality fusion with the global textual prior.

**Intermediate Fusion.** As illustrated in Fig. 3 (b), our WMF module is carefully designed for SSM-based vision-text modality fusion. Simply concatenating vision-language feature generates long sequences, resulting in exponential decay which leads to suboptimal modality fusion. To address this, we adopt a non-overlapping window partition strategy and incorporate text information to every partitioned window. Our method not only limits the sequence length for each SSM scan but also enhances cross-modal fusion capacity. Specifically, we partition $f_v^c$ into $n_{win} = H \cdot W/M$ non-overlapping windows of size $M = w_h \times w_w$, producing a window-partitioned visual feature map $f_v^{win} \in \mathbb{R}^{n_{win} \times M \times C}$. This window partitioning effectively limits each hidden state updating range, ensuring that each SSM scan only processes the tokens in a window-wise manner. We then expand the textual class token $f_t^{cls} \in \mathbb{R}^{1 \times C}$ to $f_t^{cls'} \in \mathbb{R}^{n_{win} \times 1 \times C}$ and concatenate it to each window sequence $f_v^{win}$, forming compact sequences $x_j \in \mathbb{R}^{(M+1) \times C}$ for window-wise SSM scanning with each visual patch retaining a strong connection to the textual prior. This process can be defined as follows:

$$
\begin{aligned}
x_j &= \left[ f_t^{cls'}, \; f_v^{win}[j] \right], \; j = 1, \ldots, n_{win}, \\
X &= \{x_j\}_{j=1}^{n_{win}} \in \mathbb{R}^{n_{win} \times (M+1) \times C}, \\
Y &= \mathrm{SSM}_j(X) \in \mathbb{R}^{n_{win} \times (M+1) \times C},
\end{aligned}
\tag{4}
$$

where $[\cdot, \cdot]$ denotes the concatenation and $\mathrm{SSM}(\cdot)$ refers to the selective scanning operation that captures long-range contextual information. Since each sequence length is fixed to $M + 1$, WMF prevents excessively long sequences as in full-resolution flattening, effectively mitigating the exponential decaying inherent to SSMs. Moreover, by prepending a textual class token to every window, WMF maintains a strong textual prior across all local regions, enabling robust and fine-grained modality fusion. Note that SSM is applied in parallel across all windows, so each subregion is processed concurrently. After running these sequences through SSM, we split $Y$ back into the fused visual features $f_v^{wf} \in \mathbb{R}^{n_{win} \times M \times C}$ and the updated textual class token $f_t^{fused} \in \mathbb{R}^{n_{win} \times 1 \times C}$. As a result, our WMF module delivers robust vision-text modality fusion with controlled sequence length, significantly alleviating the exponential decay.

## 2.4 Loss Function

We employ a text-to-pixel contrastive loss Wang et al. (2022) $\mathcal{L}_{con}$ to train WIMFRIS and to pull corresponding text and pixel features together while pushing apart all non-matching pairs. Let $\mathcal{P}$ and $\mathcal{N}$ be the sets of positive and negative pixel indices for a given expression. For each pixel $i$, we denote by $Z_t \in \mathbb{R}^C$ the text feature and by $Z_c^i \in \mathbb{R}^C$ the visual feature at the pixel level $i$. We define the per-pixel loss as follows:

$$
\mathcal{L}_{con}^i(Z_t, Z_c^i) = \begin{cases} -\log\big(\sigma(Z_t \cdot Z_c^i)\big), & i \in \mathcal{P}, \\ -\log\big(1 - \sigma(Z_t \cdot Z_c^i)\big), & i \in \mathcal{N}, \end{cases}
\tag{5}
$$

The overall contrastive objective averages over all labeled pixels:

$$
\mathcal{L}_{con}(Z_t, Z_c) = \frac{1}{|\mathcal{P} \cup \mathcal{N}|} \sum_{i \in \mathcal{P} \cup \mathcal{N}} \mathcal{L}_{con}^i(Z_t, Z_c^i).
\tag{6}
$$

Minimizing $\mathcal{L}_{con}$ encourages the text-pixel pairs to have high similarity while simultaneously discouraging any unrelated text–pixel alignments. We also incorporate two auxiliary losses to better supervise segmentation mask quality and cross-modal consistency. To directly optimize the mask overlap, we define the Dice loss. Let $p_i \in [0, 1]$ be the predicted probability at pixel $i$ and $g_i \in \{0, 1\}$ the corresponding ground truth mask label. Denoting **p** and **g** for the flattened prediction and the ground truth vectors of length $M$, we compute the following dice score:

$$
\mathrm{Dice}(\mathbf{p}, \mathbf{g}) = \frac{2 \sum_{i=1}^{M} p_i^{(b)} g_i^{(b)}}{\sum_{i=1}^{M} p_i^{(b)} + \sum_{i=1}^{M} g_i^{(b)} + \varepsilon},
\tag{7}
$$

where $\varepsilon = 1e - 6$ for numerical stability. The Dice loss is then computed as follows:

$$
\mathcal{L}_{dice} = 1 - \frac{1}{B} \sum_{b=1}^{B} (\mathrm{Dice}(\mathbf{p}^{(b)}, \mathbf{g}^{(b)})),
\tag{8}
$$

which encourages maximal overlap between the predicted and true masks. To align the per-window visual activations with the referring expression, we first project each of the class token outputs from our WMF module against the global textual prior. Denote by $\mathbf{L} = \left[\ell_1, \ldots, \ell_{n_{\text{win}}}\right] \in \mathbb{R}^{B \times n_{\text{win}}}$ with $\ell_j = \langle \mathbf{c}_j, \mathbf{s} \rangle$, where $\mathbf{c}_j$ is the $j$-th class token and $\mathbf{s} \in \mathbb{R}^C$ the pooled text state. In parallel, we downsample the ground truth mask $\mathbb{G}$ to the same spatial grid $(h, w)$ via nearest-neighbor interpolation and then aggregate into non-overlapping windows of size $(w_h, w_w)$ by max-pooling, yielding a binary mask vector $\mathbf{m} = \left[m_1, \ldots, m_{n_{\text{win}}}\right] \in \{0, 1\}^{n_{\text{win}}}$. Then we apply a per-window binary cross-entropy loss as alignment loss:

$$\mathcal{L}_{\text{align}} = \frac{1}{B\, n_{\text{win}}} \sum_{b=1}^{B} \text{BCE}\left(\ell^{(b)}, m^{(b)}\right), \tag{9}$$

where BCE denotes the binary cross entropy loss. Summing all components into the final training objective:

$$\mathcal{L}_{\text{total}} = \lambda_{\text{con}}\, \mathcal{L}_{\text{con}} + \lambda_{\text{dice}}\, \mathcal{L}_{\text{dice}} + \lambda_{\text{align}}\, \mathcal{L}_{\text{align}}. \tag{10}$$

This combined loss encourages accurate mask prediction, maximizes overlap, and enforces precise text-pixel alignment.

## 3 EXPERIMENTAL RESULTS

### 3.1 DATASETS

We conducted our experiments on three widely adopted referring image segmentation benchmarks for WIMFRIS. RefCOCO Kazemzadeh et al. (2014) comprises 19,994 images sourced from MSCOCO Lin et al. (2014) with 142,210 referring expressions for 50,000 object instances. The dataset is split into 120,624 images for training, 10,834 for validation, 5,657 for test A, and 5,095 for test B, respectively. Each expression averages 3.6 words, and every image depicts at least two objects. RefCOCO+ Kazemzadeh et al. (2014) contains 19,992 images and 141,564 expressions referring to 49,856 objects, divided into 120,624 training, 10,758 validation, 5,726 test A and 4,889 test B samples. By excluding absolute location terms, RefCOCO+ poses a more demanding challenge. G-Ref Yu et al. (2016) comprises 26,711 images annotated with 104,560 expressions for 54,822 objects. These expressions were collected via Amazon Mechanical Turk, average 8.4 words in length, and include more detailed spatial and appearance descriptions. We report results using both the Google and UMD splits for G-Ref.

### 3.2 IMPLEMENTATION DETAILS

We implemented two WIMFRIS variants, each paired with different DINOv2 Oquab et al. (2023) backbone. For WIMFRIS-B we employed DINOv2-B/14, and for WIMFRIS-L we used DINOv2-L/14. In both cases, the referring expressions were encoded by the CLIP Radford et al. (2021) text encoder, and all input images were resized to $448 \times 448$ pixels. For visual-linguistic modality fusion, we inserted MSA at layers 1, 3, 5, 7, 9 and 11 in WIMFRIS-B, and at layers 2, 6, 10, 14, 18 and 22 in WIMFRIS-L, respectively. The loss weights $\lambda_{\text{con}}$, $\lambda_{\text{dice}}$, and $\lambda_{\text{align}}$ are set to 0.5, 0.3, and 0.2, respectively. MTA was also applied at layers 1, 3, 5, 7, 9 and 11 of the CLIP text encoder in both WIMFRIS-B and -L configurations. Following DETRIS Huang et al. (2025), we initialized $\alpha_0$ to 0.2. We trained WIMFRIS for 50 epochs under the Adam optimizer Kingma (2014) with an initial learning rate of 1e-4 for WIMFRIS-B and 2e-4 for WIMFRIS-L, which were reduced by a factor of 10 at epoch 35. We conducted all experiments with NVIDIA RTX4090 GPUs with a batch size of 32 for WIMFRIS-B and 64 for WIMFRIS-L. We evaluated the model performance using the mean Intersection-over-Union (mIoU) metric between the predicted masks and the ground-truths following Wang et al. (2022). We leverage the task decoder from DETRIS, replacing its cross-modal neck with our WMF.

### 3.3 COMPARISON WITH STATE-OF-THE-ART METHODS

We evaluated WIMFRIS on RefCOCO, RefCOCO+ and G-Ref, as well as on the mixed RefCOCO dataset. The results, including comparisons with both full fine-tuning and PET methods, are shown

Table 2: Comparison of State-of-the-art RIS methods and the PET RIS methods on RefCOCO, RefCOCO+ and G-Ref datasets without using extra data and Mixed RefCOCO dataset, evaluated using the mIoU metric. Models marked with $^*$ are trained on the mixed RefCOCO, RefCOCO+ and G-Ref data. The best results are written in bold.

| Method | RefCOCO | | | RefCOCO+ | | | G-Ref | | | Avg |
|---|---|---|---|---|---|---|---|---|---|---|
| | val | testA | testB | val | testA | testB | val(u) | test(u) | val(g) | |
| Full Fine-tuning | | | | | | | | | | |
| ReSTR Kim et al. (2022) | 68.4 | 72.1 | 66.4 | 57.9 | 61.5 | 50.9 | 56.2 | - | 56.9 | 61.3 |
| CRIS Wang et al. (2022) | 70.5 | 73.2 | 66.1 | 62.3 | 68.1 | 53.7 | 59.9 | 60.4 | - | 63.8 |
| LAVT Yang et al. (2022) | 74.5 | 76.9 | 70.9 | 65.8 | 71.0 | 59.2 | 63.3 | 63.6 | 63.7 | 67.7 |
| CrossVLT Cho et al. (2023) | 75.5 | 77.5 | 72.7 | 67.3 | 72.0 | 60.1 | 66.2 | 62.1 | - | 69.2 |
| VPD Zhao et al. (2023) | 73.5 | - | - | 63.9 | - | - | 63.1 | - | - | 66.8 |
| ReLA Liu et al. (2023a) | 75.6 | 77.8 | 72.8 | 70.4 | 74.8 | 63.9 | 68.7 | 69.6 | 66.9 | 71.2 |
| CGFormer Tang et al. (2023) | 76.9 | 78.7 | 73.3 | 68.6 | 73.8 | 61.7 | 67.6 | 67.8 | 65.8 | 70.5 |
| LISA-7B Lai et al. (2024) | 74.9 | 79.1 | 72.3 | 65.1 | 70.8 | 58.1 | 67.9 | 70.6 | - | 69.9 |
| MagNet Chng et al. (2024) | 76.6 | 78.3 | 72.2 | 68.1 | 73.6 | 61.8 | 67.8 | 69.3 | 65.3 | 70.3 |
| ReMamber Yang et al. (2024) | 71.6 | 73.3 | 68.4 | 61.6 | 65.8 | 54.0 | 61.1 | 61.2 | - | 64.6 |
| RISCLIP-B Kim et al. (2023) | 75.7 | 78.0 | 72.5 | 69.2 | 73.5 | 60.7 | 67.6 | 68.0 | - | 70.7 |
| RISCLIP-L Kim et al. (2023) | 78.9 | 81.5 | 75.4 | 74.4 | 78.8 | 66.8 | 71.8 | 71.7 | - | 74.9 |
| Parameter-Efficient Tuning | | | | | | | | | | |
| ETRIS Xu et al. (2023) | 70.4 | 73.7 | 66.9 | 62.1 | 68.7 | 53.5 | 60.8 | 60.5 | 58.2 | 63.9 |
| BarLeRIa Wang et al. (2024b) | 72.4 | 75.9 | 68.3 | 65.0 | 70.8 | 56.9 | 63.4 | 63.8 | 61.6 | 66.5 |
| DETRIS-B Huang et al. (2025) | 76.0 | 78.2 | 73.5 | 68.9 | 74.0 | 61.5 | 67.9 | 68.1 | 65.9 | 70.4 |
| DETRIS-L Huang et al. (2025) | 77.3 | 79.0 | 75.2 | 70.8 | 75.3 | 64.7 | 69.3 | 70.2 | 67.9 | 72.2 |
| Ours-B | 77.3 | 79.2 | 74.8 | 70.2 | 75.1 | 63.5 | 69.2 | 69.4 | 67.1 | 71.8 |
| **Ours-L** | **78.2** | **79.7** | **76.3** | **71.9** | **76.2** | **67.2** | **70.4** | **71.0** | **69.3** | **73.4** |
| With Mixed Training Data | | | | | | | | | | |
| PolyFormer-L$^*$ Liu et al. (2023b) | 76.9 | 78.5 | 74.8 | 72.2 | 75.7 | 66.7 | 72.2 | 71.2 | - | 73.5 |
| UNINEXT-L$^*$ Yan et al. (2023) | 81.2 | 82.4 | 79.1 | 72.6 | 76.3 | 66.8 | 74.1 | 75.4 | - | 76.0 |
| DETRIS-L$^*$ Huang et al. (2025) | 81.0 | 81.9 | 79.0 | 75.2 | 78.6 | 70.2 | 74.6 | 75.3 | - | 77.0 |
| **Ours-L$^*$** | **81.8** | **82.8** | **79.7** | **76.6** | **80.5** | **72.5** | **75.4** | **75.9** | - | **78.2** |

in Table 2. On the standard splits (*i.e.*, RefCOCO, RefCOCO+ and G-Ref), WIMFRIS-B achieves an average IoU of 71.8, surpassing most of prior PET methods and matching all full fine-tuning methods. Scaling up to WIMFRIS-L further raises an average IoU to 73.4, setting a new state-of-the-art across all full fine-tuning and PET approaches. When trained on mixed RefCOCO datasets, WIMFRIS-L reaches the average IoU of 77.7, achieving higher performance compared to previous methods. These results underscore the ability of WIMFRIS to align and fuse visual and linguistic features robustly, enabling effective handling of complex referring open-vocabulary expressions and dense predictions compared to previous methods.

## 3.4 ABLATION STUDIES

**Effects of Window Size of the WMF Module.** Table 3 (a) reports an ablation study on the effect of the window size. The first row corresponds to naively concatenating all visual and textual tokens without window partitioning, which yields the lowest performance, achieving IoU of 75.7 on val, 77.3 on test A, and 72.6 on test B. As the window size decreases, the number of windows increases, as well as the number of enhanced global textual priors prepended. When we use the window size of $4 \times 4$, WIMFRIS achieves its optimal performance. It is worth noting that applying the window size of $2 \times 2$ degrades performance. We attribute this to the excessive fragmentation of spatial context among patches caused by the overly small window size. These results demonstrate that naively concatenating multi-modal features exposes SSMs to excessive exponential decay and weakens modality alignment, while window partitioning effectively constrains sequence length and preserves cross-modal interactions.

**Effects of our PET Strategy.** We performed an ablation study on the RefCOCO validation split to verify each component of our PET strategy and the results are shown in Table 3 (b). The results demonstrate that each proposed module, the Multi-Scale Aligner (MSA), Emphasis Parameter (EP), and Mamba Text Adapter (MTA), incrementally and effectively contributes to the RIS performance. The MSA itself provides a performance gain, validating its ability to capture multi-scale context via

Table 3: Ablation studies on the window size of the WMF module and the PET strategy components.

(a) Window size. † denotes that we simply concatenated the multi-modal tokens without window partitioning. The second smallest window size of $4 \times 4$ yields the best performance.

| Window | RefCOCO | | |
|---|---|---|---|
| Size | val | testA | testB |
| Default† | 76.0 | 77.9 | 73.4 |
| $16 \times 16$ | 76.3 | 78.5 | 73.8 |
| $8 \times 8$ | 76.8 | 78.8 | 74.4 |
| $2 \times 2$ | 76.5 | 78.3 | 74.5 |
| $4 \times 4$ | **77.3** | **79.2** | **74.8** |

(b) PET strategy. MSA, EP, and MTA refers to Multi-Scale Aligner, Emphasis Parameter, and Mamba Text Adapter, correspondingly. Params denotes the trainable parameters of the backbone model.

| MSA | EP | MTA | Params | RefCOCO | | |
|---|---|---|---|---|---|---|
| | | | | val | testA | testB |
| ✗ | ✗ | ✗ | - | 75.7 | 77.3 | 72.6 |
| ✓ | ✗ | ✗ | 0.8M | 76.4 | 78.2 | 73.6 |
| ✓ | ✓ | ✗ | 0.8M | 76.4 | 78.2 | 73.9 |
| ✗ | ✗ | ✓ | 2.2M | 76.8 | 78.7 | 74.2 |
| ✓ | ✓ | ✓ | 3.0M | **77.3** | **79.2** | **74.8** |

the RFMixer. Performance is slightly improved by the addition of EP, validating its crucial auxiliary role in adaptively weighting the contribution of each backbone stage. In addition, the MTA proves effective, underscoring the importance of its SSM-enhanced global textual prior. The full PET framework combining all three components achieves the best results, showing that our complete PET configuration is the most effective approach.

## 3.5 QUALITATIVE RESULTS

Fig. 4 presents the visualized results of five challenging scenarios: cluttered table settings, overlapped people, occluded objects behind netting, reflected animals, and densely arranged objects. Panel (a) and (b) shows the raw image and the ground truth segmentation mask, while panel (c) displays the results of DETRIS. Panel (d) and (e) shows the results of WIMFRIS. Across all challenging cases, including densely cluttered and severe occluded scenes, WIM-FRIS better captures open-vocabulary expressions and produces finer segmentation mask compared to DETRIS by effectively aligning multi-modal features. WIMFRIS successfully distinguishes the target from similar nearby objects, guided by its enhanced global textual prior.

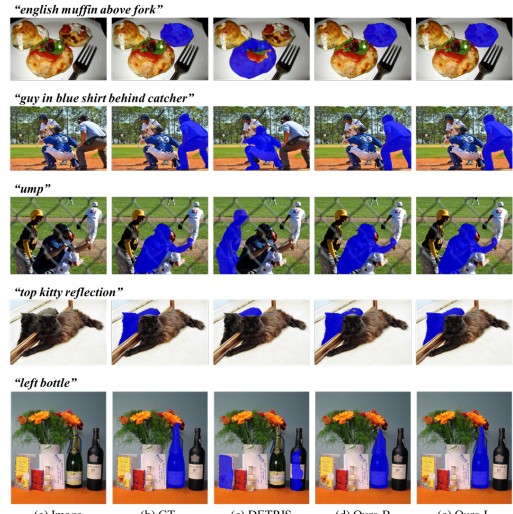

Figure 4: Qualitative results of our WIMFRIS compared with DETRIS by the input images and the referring text expressions. WIMFRIS better references the textual expressions and produces more precise segmentation masks.

## 4 CONCLUSION

We introduce WIMFRIS, a novel Referring Image Segmentation (RIS) framework featuring two parameter-efficient adapters, MTA and MSA, and a new neck architecture, the Hierarchical Mamba Fusion (HMF) block with Window Mamba Fuser (WMF) module. Our WMF module leverages a windowed design to overcome the exponential decay limitations of SSM-based fusion, while our adapters provide richer multi-modal context with minimal parameter updates. Extensive experiments on standard benchmarks validate our approach, demonstrating that WIMFRIS surpasses existing PET and full fine-tuning methods to set a new state-of-the-art for efficient and accurate RIS.

## 5 LIMITATIONS AND FUTURE WORKS

While WIMFRIS demonstrates state-of-the-art performance, we identify several promising avenues for future work. These include investigating the generalizability of our HMF block and PET strat-

egy to other dense prediction tasks to validate their versatility, exploring more efficient methods for bi-directional information flow (vision-to-text) without compromising parameter efficiency, and adapting the framework to the challenging frontier of zero-shot referring image segmentation.

## ACKNOWLEDGMENTS

This work was supported by the MSIT (Ministry of Science and ICT), Korea, under the ITRC (Information Technology Research Center) support program (IITP-2026-RS-2023-00260091) supervised by the IITP (Institute for Information & Communications Technology Planning & Evaluation) (33%), The National Research Foundation of Korea (NRF) grant funded by the Korea government (MSIT) (No. RS-2024-00414230) (33%), Development of an analog-digital mixed ultra-low power neuromorphic edge SoC (RS-2025-02263706) (33%), Samsung Electronics Co., Ltd (IO251218-14799-01), the Sogang University Grant of 202512025.01, and Institute of Information & communications Technology Planning & Evaluation (IITP) grant funded by the Korea government (MSIT) (RS-2022-00143911, AI Excellence Global Innovative Leader Education Program).

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

## A  RELATED WORKS

### A.1  PARAMETER-EFFICIENT TUNING (PET) FRAMEWORK FOR REFERRING IMAGE SEGMENTATION (RIS)

Parameter-Efficient Tuning (PET) approaches adapt pretrained models with minimal updates, reducing compute and memory overhead. Unlike full fine-tuning, which updates all weights, PET restricts updates to small adapter modules. Common strategies include adding task-specific modules or adapters Houlsby et al. (2019); Li & Liang (2021); Zhou et al. (2022), sparse parameter updates Zaken et al. (2021); Guo et al. (2020), and low-rank factorization Hu et al. (2022); Karimi Mahabadi et al. (2021); Hao et al. (2023); Liu et al. (2024a). Recently, PET has been extended to dense vision–language tasks such as Referring Image Segmentation (RIS). For example, ETRIS Xu et al. (2023) introduces a lightweight adapter called Bridger for cross-modal interaction and leverages spatial priors to refine intermediate features. BarleRIa Wang et al. (2024b) applies PET to CLIP Radford et al. (2021) encoders, addressing early fusion limitations and the lack of multi-scale

modeling. DETRIS Huang et al. (2025) employs dense interconnections between the backbone layers to improve feature propagation and mitigate alignment issues on encoders like DINO Oquab et al. (2023). These PET frameworks highlight the growing interest in efficient adaptation for dense vision-language prediction tasks.

## A.2 STATE-SPACE MODELS (SSMs) FOR VISUAL APPLICATIONS

State-Space Models (SSMs) Gu et al. (2021); Smith et al. (2022) have recently emerged as powerful architectures for capturing long-range dependencies with linear computational complexity, compared to the quadratic complexity of transformers. These models, notably exemplified by Mamba Gu & Dao (2023), have expanded from their origins in natural language processing Mehta et al. (2022) to various visual tasks Zhu et al. (2024); Ruan & Xiang (2024). In visual applications, SSMs typically address the inherent spatial and sequential dimensions of data through specialized scanning techniques. For example, VMamba Liu et al. (2024b) employs cross-scan modules to integrate spatial context effectively, overcoming the limitations of traditional state-space methods in non-causal spatial data. ReMamber Yang et al. (2024) introduces a multi-modal extension of Mamba tailored for RIS, leveraging a unique twisting mechanism that fuses textual and visual features by sequentially processing hybrid feature cubes across channel and spatial dimensions. Furthermore, recent frameworks such as MFuser Zhang & Tan (2025) utilize hybrid attention-Mamba modules to bridge the gap between vision foundation models and vision-language models, providing a robust solution for domain-generalized semantic segmentation by enhancing cross-modal alignment and spatial granularity. These advancements highlight the effectiveness and adaptability of SSMs, particularly in efficiently addressing the complex spatiotemporal dynamics of multi-modal visual tasks.

## B PRELIMINARIES

### B.1 STATE SPACE MODELS (SSMs)

SSMs model a hidden state $h(t) \in \mathbb{R}^N$ evolving over time to map an input sequence $x(t) \in \mathbb{R}$ to an output $y(t) \in \mathbb{R}$. In continuous form, they are represented by the following ordinary differential equation:

$$
\begin{aligned}
h'(t) &= Ah(t) + Bx(t), \\
y(t) &= Ch(t),
\end{aligned}
\tag{11}
$$

where $A \in \mathbb{R}^{N \times N}$ governs the internal state transitions, $B \in \mathbb{R}^{N \times 1}$ projects the input to the state space, and $C \in \mathbb{R}^{1 \times N}$ maps the hidden state to the output.

To apply such models in deep learning systems, these continuous-time equations are commonly discretized via Zero-Order Hold (ZOH) method with step size $\Delta$ yielding discrete forms of $A$ and $B$:

$$
\begin{aligned}
\overline{A} &= \exp(\Delta A), \\
\overline{B} &= (\Delta A)^{-1}(\exp(\Delta A) - I) \cdot \Delta B,
\end{aligned}
\tag{12}
$$

and the resulting discrete-time state-space equations are written as:

$$
\begin{aligned}
h_t &= \overline{A}h_{t-1} + \overline{B}x_t, \\
y_t &= Ch_t.
\end{aligned}
\tag{13}
$$

Classical SSMs employ fixed input-agnostic parameters, whereas recent advances such as Mamba Gu & Dao (2023) introduce data-dependent parameterization. Specifically, Mamba dynamically generates $B$ and $C$ from the input and adapts $\Delta$ per channel enabling input-dependent dynamics with linear-time computational complexity.

### B.2 EXPONENTIAL DECAY IN SSMs

Although discrete SSMs enable efficient linear-time sequence modeling, they suffer from a fundamental exponential decaying property that undermines long-range dependencies. Consider the

Figure 5: Additional qualitative results on RefCOCO dataset.

discrete recurrence:

$$h_t = \overline{A}\, h_{t-1} + \overline{B}\, x_t, \qquad h_0 = 0, \tag{14}$$

where $\overline{A} \in \mathbb{R}^{N \times N}$ and $\overline{B} \in \mathbb{R}^{N \times 1}$. Unrolling over $t$ steps gives the closed-form

$$h_t \;=\; \sum_{k=1}^{t} \overline{A}^{\,t-k}\, \overline{B}\, x_k. \tag{15}$$

For simplicity, we assume an induced matrix norm so that whenever the spectral radius (*i.e.,* the maximum absolute value of the eigenvalues) of $\overline{A}$ is below one, there exist constants $M > 0$ and $\lambda \in (0,1)$ satisfying $\|\overline{A}^n\| \leq M\lambda^n$ for all $n$, and we further assume a zero initial state $h_0 = 0$ with fixed $\overline{A}$ and $\overline{B}$ throughout the sequence. Taking norms and using submultiplicativity (*i.e.,* $\|XY\| \leq \|X\|\|Y\|$), we obtain

$$\|h_t\| \;\leq\; \sum_{k=1}^{t} \|\overline{A}^{\,t-k}\| \, \|\overline{B}\| \, \|x_k\|. \tag{16}$$

If $\rho(\overline{A}) < 1$ (as ensured by $\overline{A} = \exp(\Delta A)$ with $\Delta > 0$), then there exist constants $M > 0$ and $\lambda \in (0,1)$ such that

$$\|\overline{A}^n\| \;\leq\; M\,\lambda^n, \quad \forall n \geq 0. \tag{17}$$

Substituting yields

$$\|h_t\| \;\leq\; M\,\|\overline{B}\| \sum_{k=1}^{t} \lambda^{\,t-k}\,\|x_k\|. \tag{18}$$

Because $\lambda^{\,t-k}$ decays geometrically as the gap $t - k$ grows, inputs $x_k$ from even moderately distant timesteps contribute only a tiny fraction to $h_t$. Specifically, the information from past inputs decays at an exponential rate. In practice, this decay means that early tokens, whether visual patches or words in a referring expression, are effectively invisible to later states, unless very large weights or additional architectural mechanisms are introduced. Mitigating this exponential attenuation is therefore essential for tasks requiring strong long-range and cross-modal integration.

## C    ADDITIONAL QUALITATIVE RESULTS

In addition to the qualitative results in the main paper, we further display more RefCOCO Kazemzadeh et al. (2014); Yu et al. (2016) qualitative results on various referring expressions across diverse scenes in Fig. 5. The results demonstrate that WIMFRIS effectively identifies and segments target objects even in challenging scenarios such as occlusion in the first row, complex backgrounds in the second row, and multiple similar instances in the sixth and seventh rows. Notably, in scenes with fine-grained distinctions, such as differentiating among people in lab coats in the third row or distinguishing between similarly coherent masks aligned with the referring expressions. These results highlight the robustness of our WIMFRIS and its capacity to localize referred objects with high fidelity, validating its superiority in understanding nuanced textual cues and resolving ambiguities in cluttered visual environments.

## D    ADDITIONAL ABLATION STUDIES

**Effects of Kernel Size of the RFMixer.** We conducted an ablation study on multi-branch depth-wise strip convolutions in the RFMixer to determine the optimal kernel sizes, as summarized in Table 4.

Table 4: Ablation study on the kernel sizes of the convolutional branches in the RFMixer. Using kernel sizes of 3, 5, and 7 in the RFMixer branches yields the best performance.

| Kernel Size | Params (M) | RefCOCO | | |
|---|---|---|---|---|
| | | val | testA | testB |
| 1-1-1 | 3.02 | 76.7 | 78.9 | 74.7 |
| 3-3-3 | 3.03 | 76.9 | 79.2 | 74.6 |
| 5-5-5 | 3.04 | 77.0 | 79.1 | 74.6 |
| 7-7-7 | 3.05 | 77.0 | 79.0 | 74.3 |
| 1-3-5 | 3.03 | 77.2 | 79.0 | 74.8 |
| 3-5-7 | 3.04 | **77.3** | **79.2** | **74.8** |

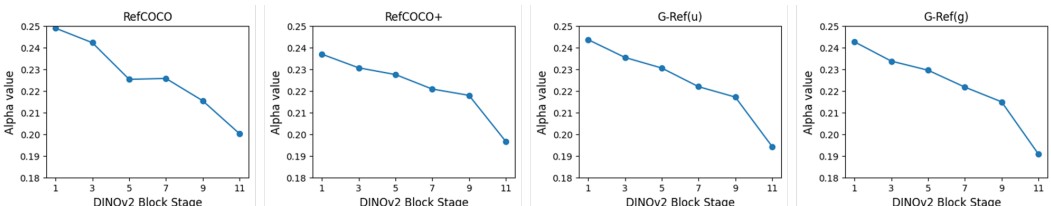

Figure 6: Visualization of the learned $\alpha$ values for the emphasis parameter. The plots show a consistent downward trend across all datasets, indicating that the model learns to apply stronger fine-tuning to the early, low-level feature layers of the DINOv2 backbone and progressively less to the deeper, high-level semantic layers.

Here, a kernel configuration of $a - b - c$ denotes three convolutional branches with depth-wise strip kernels of size $a$, $b$, and $c$, respectively. Experimental results show that the $3 - 5 - 7$ kernel sizes delivers the highest IoUs. These results validate that leveraging multiple receptive fields via the RFMixer effectively captures rich multi-scale contextual information, yielding fine-grained visual features for dense prediction for referring image segmentation.

**Emphasis Parameter Analysis.** We visualize the learned $\alpha$ values of the emphasis parameters after training WIMFRIS-B to analyze the layer-wise contribution of our Multi-Scale Aligner (MSA). As shown in Fig. 6, there is a consistent downward trend in $\alpha$ values across all datasets as the DINOv2 block stage deepens. The values are highest in the initial layers and progressively decrease towards the final layer. This indicates that our model adaptively applies more intensive fine-tuning to the early layers, which capture more generic, low-level visual features such as colors or edges. Conversely, the high-level semantic representations from the deeper layers of the self-supervised DINOv2 backbone are already robust and require less task-specific adaptation. This result validates the efficacy of our learnable emphasis parameters, which enable the model to dynamically allocate tuning strength where it is most needed for effective vision-language alignment.

**Adapter Placement Analysis.** We conducted an ablation study comparing our default adapter distribution against depth-specific configurations to investigate the optimal placement strategy for our PET adapters. We grouped adapters into low-level (layers 0-5), mid-level (layers 3-8), and high-level (layers 6-11) concentrations while maintaining a constant parameter budget, and compared them with our default placement configuration (layers 1, 3, 5, 7, 9, 11). As presented in Table 5, the default configuration consistently yields superior performance compared to all concentrated configurations. This validates that the comprehensive adaptation of hierarchical representations is the most effective, ranging from low-level visual details in shallow layers to high-level semantic abstractions in deep layers. Concentrating adapters at a specific depth restricts the model to fine-tune this full spectrum of information. Furthermore, this distributed placement validates the synergy with our Emphasis Parameters (EP), as it provides diverse multi-level inputs that allow the EP module to dynamically optimize the contribution of each layer for effective vision-language alignment.

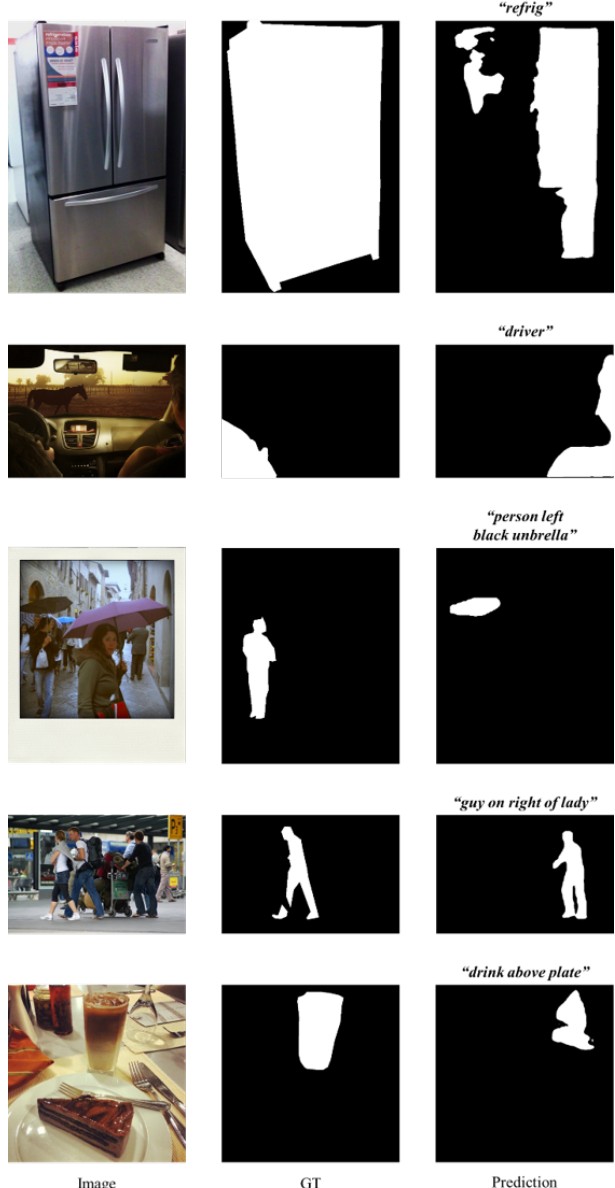

Figure 7: Notable failure cases. Erroneous prediction masks are produced when the target object is excessively large (first row), when the context necessary to specify the target object is not directly provided or ambiguous within the referring expression (second and third row), or when the required specifying context is spatially distant from the target object (fourth and fifth rows).

# E    NOTABLE FAILURE CASES

In this section, we present failure case analysis of our work for future works. As shown in the first row in Fig. 7, erroneous masks occurred when the target object is excessively large, spanning multiple windows. Since WMF partitions the visual features into non-overlapping windows and processes them in parallel to mitigate exponential decay, the model identifies the object as fragmented patches rather than a unified one. In the second and third rows in Fig. 7, WIMFRIS incorrectly segmented the target object when the referring expression is ambiguous or lacks specific context, since each window independently aligns its local visual features with the shared global textual prior in our HMF neck. In the fourth and fifth rows in Fig. 7, when the visual context required to disambiguate the target is spatially distant and located in a different window, WIMFRIS yielded some

Table 5: Ablation study on the Impact of PET adapter placement strategies. Default $[1, 3, 5, 7, 9, 11]$ configuration yields the best performance. Note that the layer number is zero-indexed.

| Adapter Placement | RefCOCO | | |
|---|---|---|---|
| | val | testA | testB |
| $[0, 1, 2, 3, 4, 5]$ | 76.6 | 78.4 | 74.1 |
| $[3, 4, 5, 6, 7, 8]$ | 76.8 | 78.3 | 74.1 |
| $[6, 7, 8, 9, 10, 11]$ | 76.6 | 78.6 | 73.8 |
| $[1, 3, 5, 7, 9, 11]$ | **77.3** | **79.2** | **74.8** |

failed prediction. This is due to the spatial isolation inherent in window partitioning where the window containing the target object cannot directly access the visual features of the context object in a separate window. Given the aforementioned limitations, future works are encouraged to investigate carefully designed fusion module capable of bridging the spatial isolation of window partitioning, thereby enabling a better understanding of large objects and distant visual contexts.

## F  STATEMENT ON LARGE LANGUAGE MODELS USAGE

In the interest of full transparency and adherence to the ICLR 2026 policy, we report that a large language model (LLM) was leveraged to assist in the refinement of this manuscript.

**Scope of Use.** The LLM's function was exclusively that of a writing assistant. Its usage was limited to:

- Correcting grammatical errors and refining word choices to improve precision and conciseness.
- Restructuring sentences to enhance clarity and logical flow.
- Assisting with the positioning of tables and figures within the LaTeX environment.

