# OpenReview forum: "WIMFRIS: WIndow Mamba Fusion and Parameter Efficient Tuning for Referring Image Segmentation"
_ICLR.cc/2026/Conference — ICLR 2026 Poster_

### Official Review · Reviewer_UdVq · 2025-10-28

**Soundness:** 3
**Presentation:** 3
**Contribution:** 3
**Rating:** 6
**Confidence:** 5

**Summary:**

WIMFRIS introduces a neck-heavy, parameter-efficient RIS framework that aggregates multi-scale DINOv2 features, fuses them with CLIP text via a windowed Mamba block, and adaptively re-weights each stage, setting new SOTA mIoU on RefCOCO/+/g with < 3 % trainable params.

**Strengths:**

1. First to plug a windowed SSM neck (WMF) into RIS; mitigates exponential decay of vanilla Mamba.
2. Learnable emphasis per stage is simple yet novel for PET.
3. Exhaustive ablations: window size, kernel configs, PET modules all explored.
4. Plug-in HMF boosts ETRIS & DETRIS (Table 1), proving generic utility.

**Weaknesses:**

1. All results are fine-tuned; real-world deployment often lacks target-domain labels.
2. WMF prepends text to windows, but vision never feeds back to text; may miss visual disambiguation cues.
3. Parameter efficiency ≠ inference speed; window partitioning + SSM may hurt parallelism.

**Questions:**

See weakness

---

> ### Author Response · Authors · 2025-11-21
> **Response to Reviewer UdVq**
>
> We sincerely appreciate you taking the time to review this article. The answers to each question are provided below.
>
> **About real-world deployment without target-domain labels**
>
> Thank you for the important point regarding real-world deployment. We analyze the performance on unseen classes by conducting an additional experiment following the dataset splitting strategy of CGFormer [3], where we divided the RefCOCO val and test sets into seen and unseen splits. The results below show that our WIMFRIS achieves superior generalization performance on the unseen classes compared to existing state-of-the-art methods. This validates that our architecture is effective not only for standard fine-tuning but also for real-world deployment scenarios where target-domain labels are scarce.
>
> | Method | RefCOCO Val (Seen) | RefCOCO Val (Unseen) | RefCOCO Test (Seen) | RefCOCO Test (Unseen) |
> | :--- | :---: | :---: | :---: | :---: |
> | CRIS [1] | 68.7 | 52.8 | 52.8 | 52.7 |
> | LAVT [2] | 73.1 | 61.4 | 72.3 | 57.7 |
> | CGFormer [3] | 75.5 | 63.2 | 74.6 | 59.0 |
> | **Ours-B** | **76.7** | **65.8** | **76.1** | **62.6** |
>
>
> **About text-to-vision prepending**
>
> Thank you for this insightful comment regarding the direction of information flow. Our WMF module indeed primarily implements a text-to-vision flow. Following established approaches such as LAVT [2] and RISCLIP [4], the cross-attention mechanisms in backbone encoder handle the initial vision-to-text information flow, with visual features serving as the query, and the textual features serving as the key and value, ensuring that the textual features are already vision-aware and contextualized before reaching the HMF neck. Consequently, the specific role of our HMF neck is to effectively broadcast this pre-aligned global textual information into the local visual windows to resolve spatial ambiguities in the partitioned features. We believe that this design optimizes the overall capacity of effective final pixel-level predictions while preventing the loss of visual disambiguation cues.
>
> **About efficiency and parallelism of window partitioning**
>
> Thank you for your insightful comment regarding a detailed efficiency analysis. We measured the GFLOPs, FPS, total inference time, and total number of trainable parameters on a single 24G RTX4090 GPU. (Note that ETRIS [5] uses CLIP [7] as its vision encoder, while both DETRIS [6] and our WIMFRIS use DINOv2 [8].) The results below show that even though WIMFRIS has more total trainable parameters than DETRIS, it achieves significantly better performance with comparable computational cost (GFLOPs) and inference speed (FPS). We believe this result demonstrates the efficiency of our proposed WMF module. This efficiency is achieved by architectural design, as the WMF is structured to process all windows in parallel using Mamba's efficient SSM scanning mechanism. Specifically, our WMF module first partitions the input visual feature map into $n_{win}$ non-overlapping windows, converting a single, long sequence into $n_{win}$ short sequences of length $M$. We then expand and prepend the global textual class token to each of these windows, creating a tensor of shape $(B, n_{win}, M+1, C)$. The key to our parallel implementation is reshaping this tensor to $(B \cdot n_{win}, M+1, C)$. This merges the window dimension into the batch dimension, allowing the GPU to process the Mamba SSM scan as a batch of $(B \cdot n_{win})$ items, each with a very short sequence length of $(M+1)$. This design ensures all $(B \cdot n_{win})$ fusions between the local visual windows and the global text token occur simultaneously and in parallel, making the architecture highly efficient.
>
> | Method | GFLOPs | FPS | Total Inference Time (sec) | Total Trainable Params (M) | Performance (RefCOCO-val) |
> | :--- | :---: | :---: | :---: | :---: | :---: |
> | ETRIS [5] | 64.85 | 15.81 | 241 | 25.9 | 70.4 |
> | DETRIS [6] | 128.67 | 10.87 | 350 | 25.5 | 76.0 |
> | **Ours-B** | **128.56** | **10.63** | **358** | **27.9** | **77.3** |

---

> > ### Author Response · Authors · 2025-11-21
> > **Response to Reviewer UdVq**
> >
> > References
> >
> > [1] WANG, Zhaoqing, et al. Cris: Clip-driven referring image segmentation. In: Proceedings of the IEEE/CVF conference on computer vision and pattern recognition. 2022. p. 11686-11695.
> >
> > [2] YANG, Zhao, et al. Lavt: Language-aware vision transformer for referring image segmentation. In: Proceedings of the IEEE/CVF conference on computer vision and pattern recognition. 2022. p. 18155-18165.
> >
> > [3] TANG, Jiajin, et al. Contrastive grouping with transformer for referring image segmentation. In: Proceedings of the IEEE/CVF conference on computer vision and pattern recognition. 2023. p. 23570-23580.
> >
> > [4] KIM, Seoyeon, et al. Extending clip’s image-text alignment to referring image segmentation. In: Proceedings of the 2024 Conference of the North American Chapter of the Association for Computational Linguistics: Human Language Technologies (Volume 1: Long Papers). 2024. p. 4611-4628.
> >
> > [5] XU, Zunnan, et al. Bridging vision and language encoders: Parameter-efficient tuning for referring image segmentation. In: Proceedings of the IEEE/CVF international conference on computer vision. 2023. p. 17503-17512.
> >
> > [6] HUANG, Jiaqi, et al. Densely connected parameter-efficient tuning for referring image segmentation. In: Proceedings of the AAAI Conference on Artificial Intelligence. 2025. p. 3653-3661.
> >
> > [7] RADFORD, Alec, et al. Learning transferable visual models from natural language supervision. In: International conference on machine learning. PmLR, 2021. p. 8748-8763.
> >
> > [8] OQUAB, Maxime, et al. Dinov2: Learning robust visual features without supervision. arXiv preprint arXiv:2304.07193, 2023.

---

> > > ### Comment · Reviewer_UdVq · 2025-11-27
> > >
> > > Thanks for the new experiments supporting your claims. I will keep my score.

---

### Official Review · Reviewer_CZEK · 2025-10-29

**Soundness:** 2
**Presentation:** 1
**Contribution:** 2
**Rating:** 4
**Confidence:** 3

**Summary:**

This paper proposes a novel parameter-efficient tuning (PET) method named WIMFRIS for referring image segmentation. In contrast to existing PET methods that primarily focus on layer-wise feature alignment and are struggle to aggregate multi-scale features, the proposed approach introduces a simple yet effective neck architecture based on the Mamba module. WIMFRIS achieves state-of-the-art performance on standard RIS benchmarks, demonstrating both efficiency and strong segmentation capability.

**Strengths:**

- The paper proposes a new efficient parameter-efficient tuning (PET)–based referring image segmentation (RIS) approach named WIMFRIS.
- The proposed algorithm enhances efficiency by replacing conventional blocks with an HMF block that actively leverages the Mamba architecture. In addition, it introduces several novel components—an SSM-based MTA, an MSA robust to multiple receptive fields, and an RFMixer—which together contribute to more precise vision-language fusion.
- The method achieves state-of-the-art performance on popular RIS benchmarks, demonstrating both effectiveness and robustness.

**Weaknesses:**

- Structural Issues in Writing
   - In the Abstract, abbreviations such as HMF and WMF appear without their full names or descriptions, making it difficult for readers to understand them.
   - Figure 1 lacks an explanation of the HMF module, requiring readers to infer that WMF is a sub-module of HMF only from context.
- #Params of PET and Performance Comparison
   - When comparing with existing PET methods, it would be fair to keep the number of PET parameters (#params) consistent across models. According to Table 1, when DINOv2-B/14 is used as the vision encoder, the proposed method shows only a slight improvement in performance compared to DETRIS, even though it uses more parameters. This raises concerns that the effectiveness of WIMFRIS may not be scalable.
- Limited Novelty
   - The paper proposes several modules (e.g., WMF, HMF, MSA, MTA), but the architectural novelty of each component seems limited. For instance, the HMF module appears to replace multiple cross-attention layers with a more efficient Mamba-based structure, but the use of Mamba itself is not novel. Similarly, the MSA and RFMixer are designed to handle multiple receptive fields, but this concept is not entirely new.
   - The paper would benefit from additional discussion or evidence to substantiate the novelty of these architectural contributions.
- Lack of Ablation Studies
   - As mentioned above, the paper lacks experiments that demonstrate the effectiveness and novelty of the proposed modules. For example, it would strengthen the work to include comparisons between MSA/RFMixer and baseline or vanilla methods for handling multiple receptive fields.
   - Table 3-(a) appears more like an engineering-oriented study rather than one providing clear scientific insight.

**Questions:**

Please provide your responses with reference to the weaknesses mentioned above.

---

> ### Author Response · Authors · 2025-11-21
> **Response to Reviewer CZEK**
>
> We sincerely appreciate you taking the time to review this article. The answers to each question are provided below.
>
> **Structural issues in writing**
>
> We sincerely appreciate you pointing out these important oversights, which are crucial for the paper’s readability. We have made the following corrections. First, the abstract has been revised to ensure that all abbreviations, such as HMF and WMF, are fully defined when they are first introduced. Second, we have updated the caption to include a clear explanation of the HMF module and its relationship with WMF, ensuring readers no longer need to infer this information from context.
>
> **About fair number of parameters and performance comparison**
>
> Thank you for this crucial point. We completely agree that a fair comparison, especially one that keeps the number of PET parameters consistent, is essential for validating our model’s effectiveness and scalability. To provide the most direct and fair comparison, we conducted an additional experiment specifically to isolate the contribution of our neck. In the experiment, we fixed the PET parameters of DETRIS [4] and our WIMFRIS to 1.4M, identical to ETRIS [3], and set the text encoder to CLIP [1] and the vision encoder to DINOv2 [2]. The results below show that leveraging our HMF neck yields a consistent performance improvement on the RefCOCO val split. Since the PET framework and parameter count are held consistent in this specific comparison, this result validates that the performance gain is not due to an increased parameter count but is a direct result of our HMF’s architectural design. We believe that this confirms the effectiveness and scalability of our proposed neck module.
>
> | Text enc. | Vision enc. | PET method | # Params (M) | Neck | RefCOCO Val | RefCOCO TestA | RefCOCO TestB |
> | :--- | :--- | :--- | :---: | :---: | :---: | :---: | :---: |
> | CLIP-ViT-B/16 [1] | DINOv2-B/14 [2] | ETRIS [3] | 1.4 | X | 72.2 | 73.9 | 70.8 |
> | | | | | ETRIS | 74.5 | 76.5 | 72.9 |
> | | | | | **Ours** | **75.7** | **77.5** | **73.3** |
> | | | DETRIS [4] | 1.4 | X | 74.3 | 75.8 | 70.8 |
> | | | | | DETRIS | 75.8 | 77.7 | 72.9 |
> | | | | | **Ours** | **76.4** | **78.3** | **73.6** |
> | | | **Ours** | 1.4 | **Ours** | **77.2** | **78.9** | **74.3** |
>
> **Limited novelty**
>
> Thank you for your valuable feedback. We appreciate the opportunity to substantiate the novelty of our architectural contributions. We agree that components like Mamba and the concept of multi-receptive fields are existing tools. However, we believe that our novelty lies in the specific architectural designs that solve key bottlenecks, that our window-partitioning strategy mitigates exponential decaying problem of SSMs, and our PET modules enable efficient fine-tuning. Our contribution is not the mere adaptation of Mamba itself, but our WMF module. A naïve Mamba replacement for cross-attention would flatten multi-modal features into a single, long sequence, which would suffer from the exponential decaying problem inherent to SSMs, which we explicitly aimed to solve. Our WMF’s novelty is its specific window-partitioning strategy. It creates short, parallel sequences by prepending a shared global textual prior to each window sequence. This novel design is not a simple replacement but a specific architecture that effectively mitigates exponential decay and ensures every local visual region interacts directly with the global textual context. Similarly, regarding the MSA and RFMixer, while the concept of multi-receptive fields exists, our novelty lies in its specific, efficient implementation as a PET adapter. The RFMixer proposes a lightweight design using parallel branches of depth-wise strip convolutions, a specific and efficient choice validated by our ablation studies (Table 3(b) and Table 4).

---

> > ### Author Response · Authors · 2025-11-21
> > **Response to Reviewer CZEK**
> >
> > **Lack of ablation studies**
> >
> > Thank you for the suggestion. We agree that demonstrating the effectiveness of each module is critical. To address your concern regarding the MSA and RFMixer, we have reattached Table 4 from the Appendix below for your convenience. We apologize for relegating the critical analysis to the Appendix, which may have obscured its importance. This experiment provides the comparison by evaluating various kernel configurations, effectively comparing our multi-scale approach against single scale (vanila i.e., 1-1-1, 3-3-3, and 5-5-5 kernel sizes) baselines. The results demonstrate that our proposed parallel strip convolution (3-5-7 configuration) outperforms single-scale or suboptimal configurations. This validates that leveraging multiple receptive fields is not just an engineering choice but a structural necessity for capturing rich multi-scale contextual information in this task. However, if we have misunderstood your suggestion, we are more than willing to conduct further experiments to provide that comparison.
> > Regarding Table 3(a), we respectfully clarify that the primary purpose of Table 3(a) is to provide scientific insight into the behavior of SSMs, rather than merely finding engineering-oriented hyperparameters. The most critical finding in Table 3(a) is the failure of the ‘default’ baseline, which naively concatenates features without window partitioning. This result empirically verifies our core hypothesis that standard SSMs suffer from exponential information decay when processing long multi-modal sequences. In addition, by comparing this baseline with our window-based configurations, we scientifically establish that window partitioning is the key mechanism to mitigate this information decay. While finding the specific window size involves empirical optimization, the experiment itself validates our window-partitioning strategy against the standard Mamba approach. We believe these analyses provide the scientific evidence required to substantiate the novelty and effectiveness of our architectural contributions. However, if you have a different analytical approach in mind that could provide deeper scientific insight beyond this empirical study, we would be more than willing to conduct those additional experiments.
> >
> > | Kernel Size | Params (M) | RefCOCO val | RefCOCO testA | RefCOCO testB |
> > | :--- | :---: | :---: | :---: | :---: |
> > | 1-1-1 | 3.02 | 76.7 | 78.9 | 74.7 |
> > | 3-3-3 | 3.03 | 76.9 | 79.2 | 74.6 |
> > | 5-5-5 | 3.04 | 77.0 | 79.1 | 74.6 |
> > | 7-7-7 | 3.05 | 77.0 | 79.0 | 74.3 |
> > | 1-3-5 | 3.03 | 77.2 | 79.0 | 74.8 |
> > | 3-5-7 | 3.04 | **77.3** | **79.2** | **74.8** |
> >
> > References
> >
> > [1] RADFORD, Alec, et al. Learning transferable visual models from natural language supervision. In: International conference on machine learning. PmLR, 2021. p. 8748-8763.
> >
> > [2] OQUAB, Maxime, et al. Dinov2: Learning robust visual features without supervision. arXiv preprint arXiv:2304.07193, 2023.
> >
> > [3] XU, Zunnan, et al. Bridging vision and language encoders: Parameter-efficient tuning for referring image segmentation. In: Proceedings of the IEEE/CVF international conference on computer vision. 2023. p. 17503-17512.
> >
> > [4] HUANG, Jiaqi, et al. Densely connected parameter-efficient tuning for referring image segmentation. In: Proceedings of the AAAI Conference on Artificial Intelligence. 2025. p. 3653-3661.

---

> > > ### Comment · Reviewer_CZEK · 2025-11-27
> > >
> > > Thanks for the new experiments supporting your claims. I will raise my rating.

---

### Official Review · Reviewer_WGQk · 2025-10-30

**Soundness:** 3
**Presentation:** 3
**Contribution:** 4
**Rating:** 6
**Confidence:** 5

**Summary:**

This paper introduces WIMFRIS, a framework for Referring Image Segmentationthat focuses on both a novel intermediate fusion neck architecture (the Hierarchical Mamba Fusion, or HMF, block) and a parameter-efficient tuning strategy. The HMF block leverages a Window Mamba Fuser module to effectively aggregate and fuse multi-scale vision and language features, using window partitioning to tackle the exponential decay in information typical of state-space models. The PET strategy employs adapters to efficiently align textual and visual representations and a learnable stage-wise emphasis mechanism. Extensive experiments are conducted on major RIS benchmarks, demonstrating state-of-the-art results for WIMFRIS compared to both PET-based and full fine-tuning methods.

**Strengths:**

- WIMFRIS achieves state-of-the-art or highly competitive performance across all standard RIS benchmarks (RefCOCO, RefCOCO+, G-Ref), outperforming previous parameter-efficient and full-tuning baselines. Table 2 clearly demonstrates these gains, including mixed-data setups.
- Multiple ablation tables systematically dissect the contributions of each module and architectural choice.
- The schematic diagrams provide clear breakdowns of the model pipeline, supporting the text’s descriptions of modular design and the flow of visual and textual feature processing. The visualizations  offer compelling qualitative evidence for improved segmentation, especially in challenging situations (e.g., clutter, occlusion).
- The paper carefully characterizes the underlying exponential decay issue in SSM-based fusion, and the model’s windowed approach is well justified both mathematically and empirically.
- WIMFRIS demonstrates competitive results while tuning a very small fraction of backbone parameters, highlighting the value for practical deployment.
- The explicit, detailed description of contrastive, dice, and alignment losses (and their weighting) makes reproduction feasible and testable.

**Weaknesses:**

- While MSA adapters and MTA are described and visualized in Figure 2, the specific methodology for choosing insertion layers for adapters in different backbones is only loosely justified. There is a missed opportunity for a principled, possibly automated or analytical policy for placement, and no ablation on layer choice is provided.
- Although Table 3 (a) explores performance trade-offs for window size, the choice of optimal $4 \times 4$ is only empirically justified. There is little theoretical or dataset-specific reasoning for why this size generalizes, and exploring task- or scale-adaptive policies would strengthen claims of robustness.
- There are several grammatical errors and awkward phrasings, as well as the use of slightly non-standard abbreviations in the tables (e.g., "vol", "m/s/6", "m/sfI" in Table 1), which may disrupt readability and hinder quick assimilation for a broad audience.

**Questions:**

- Can the authors provide a rationale for the placement of PET adapters (MSA, MTA) at specific depths in the vision/text backbone? Have they considered or tested more adaptive/learned strategies for insertion, and can they provide ablations or guidelines for optimal selection?

- How is the concatenation between text class tokens and visual patch windows actually handled in practice (e.g., with respect to normalization, possible channel mismatch, and possible overfitting due to repetitive text tokens)? Would normalization before SSM scans improve performance or stability?

- Have the authors empirically measured the actual decay rate of long-range dependencies for varying window sizes in SSM, and if so, can those be reported? Is the optimal window size truly dataset/task dependent?

- Are there notable scenarios where the windowed approach harms segmentation accuracy, e.g., in very small or oddly-shaped object instances, or when referring expressions are ambiguous or highly context-dependent?

- Will the complete code (including all adapter implementations and ablation regimes) be released for reproducibility, and if so, under what license and conditions?

---

> ### Author Response · Authors · 2025-11-21
> **Response to Reviewer WGQk**
>
> We sincerely appreciate you taking the time to review this article. The answers to each question are provided below.
>
> **Rationale for the place of PET adapters at specific depths in the vision/text backbone**
>
> Thank you for your insightful comments and for pointing out this crucial issue. To explain using our WIMFRIS-B model as a baseline, we utilize six PET adapters (for both MSA and MTA) to prevent an excessive increase in the number of trainable parameters. If the adapters are concentrated at a specific depth, the model cannot effectively fine-tune the diverse levels of semantic information in a task-specific manner, since backbones encode hierarchical features. In addition, this placement can maximize the efficacy of our emphasis parameters (EP), which dynamically weight contributions from diverse inputs from varying layer depths. To directly address your concerns regarding this placement, we have conducted additional experiments. Specifically, we compared the performance of our default setting (applying adapters to layers 1, 3, 5, 7, 9, and 11) against configurations where adapters were concentrated in low-level, mid-level, or high-level layers. The results are presented in the table below. (Please note that the layers are zero-indexed.)
>
> | Adapter Placement | RefCOCO Val | RefCOCO TestA | RefCOCO TestB |
> | :--- | :---: | :---: | :---: |
> | [0,1,2,3,4,5] | 76.6 | 78.4 | 74.1 |
> | [3,4,5,6,7,8] | 76.8 | 78.3 | 74.1 |
> | [6,7,8,9,10,11] | 76.6 | 78.6 | 73.8 |
> | **[1,3,5,7,9,11]** | **77.3** | **79.2** | **74.8** |
>
> The results show that our default setting (layers 1, 3, 5, 7, 9, and 11) yields the best performance, demonstrating that tuning semantic information across all levels (low, mid, and high) is the most optimal.
>
> **Optical window size justification**
>
> Thank you for the insightful questions regarding the justification for the optimal window size. To address your point, we conducted additional experiments of optimal window size on RefCOCO+ and G-Ref datasets. The results demonstrate that the 4x4 window size consistently achieves optimal performance across all RefCOCO family datasets, validating it as the optimal choice. We believe that the optimal window size is dataset/task dependent. Our results demonstrate a clear trade-off. While an even smaller window size, such as 2 x 2, shortens the sequence length even further, our experiments show it consistently degrades performance compared to 4 x 4. We attribute this to the excessive fragmentation of visual context. When the windows become too small, meaningful spatial information is broken apart, making it difficult for the model to learn. Therefore, the optimal window size represents the best balance between keeping sequences short to mitigate SSM decay and preserving enough local visual context to be meaningful. We believe this optimal balance would likely vary depending on factors like the dataset’s native image resolution or the nature of the prediction task such as pixel-level prediction.
>
> **About grammatical errors and awkward phrasings**
>
> We sincerely appreciate your valuable feedback. We have carefully reviewed the entire manuscript and addressed the grammatical errors and awkward phrasings.
>
> **Notable scenarios where the windowed approach harms segmentation accuracy or when referring expressions are ambiguous or highly context-dependent**
>
> Thank you for highlighting this important aspect. As you suggested, there are notable scenarios where our window-based approach encounters difficulties. Specifically, we observed suboptimal segmentation masks in cases where the referring expression is ambiguous or lacks specific context, since each window independently aligns its local visual features with the shared global textual prior in our HMF neck. In addition, when the visual context required to disambiguate the target is spatially distant and located in a different window, WIMFRIS yielded some failed prediction. This is due to the spatial isolation inherent in window partitioning where the window containing the target object cannot directly access the visual features of the context object in a separate window. Another failure case occurred when large objects that span multiple windows, where the model sometimes predicts only fragmented parts of the object. We fully recognize that analyzing these failure cases is vital for guiding future research. We have added a detailed discussion of these scenarios to the Appendix.
>
> **About complete code release**
>
> Thank you for your comment regarding reproducibility. We are fully committed to this principle. The complete code will certainly be released under an MIT license, making it fully accessible to the entire research community.

---

> > ### Author Response · Authors · 2025-11-21
> > **Response to Reviewer WGQk**
> >
> > **Clarification about concatenation between text class tokens and visual patch windows in practice, and layer normalization before SSM**
> >
> > Thank you for this meaningful question. The core objective of our WMF module is to efficiently fuse the global information from the textual CLS token with the local information from the image feature map. To achieve this, we first take the full-resolution input feature map $x \in (B, C, H, W)$ and apply a rearrange operation to partition it into $(g_h \cdot g_w)$ small windows. (Prior to this, to prevent a possible channel mismatch, we apply a Linearization layer to the textual class token and a $1 \times 1$ convolution to the visual feature map, respectively.) This initial transformation reshapes the tensor to $(B, n_{win}, M, C)$, where $n_{win}$ is the total number of windows $(g_h \cdot g_w)$ and $M$ is the number of pixels within a window $(w_h \cdot w_w)$. This partitioning is a crucial step to reduce computational complexity, as it converts one long sequence of length $(H \cdot W)$ into $n_{win}$ short sequences. Next, we inject the textual information into each window. The textual class token is expanded from $(1, B, C)$ to $(B, n_{win}, 1, C)$, effectively replicating it for all $n_{win}$ windows. This expanded textual class token is then concatenated with the transformed visual tensor $(B, n_{win}, M, C)$ along the sequence dimension (dim=2). This results in a tensor of shape $(B, n_{win}, M+1, C)$. A subsequent rearrange operation is the key to our parallel processing. The $(B, n_{win}, M+1, C)$ tensor is reshaped into $(B \cdot n_{win}, M+1, C)$. This transformation merges the $n_{win}$ dimension with the batch dimension. This allows the GPU to perceive the operation as a single batch job with a very large batch size $(B \cdot n_{win})$ but a very short sequence length $(M+1)$. The SSM scanning operation via Mamba is then performed on this $(B \cdot n_{win}, M+1, C)$ tensor. This structure is efficient for GPU computation, as the entire $(B \cdot n_{win})$ batch size is fully parallelized. Each window is treated as an independent batch instance, allowing $(B \cdot n_{win})$ separate fusions between the text token and the local window pixels to occur simultaneously. After the scan, the output token is reassembled from $(B \cdot n_{win}, M+1, C)$ back to $(B, n_{win}, M+1, C)$ and finally rearranged into the fused feature map with the original image shape $(B, C, H, W)$. The design choice of prepending the same textual class token to every window is a core element of our architecture. It ensures that all windows are consistently processed within the context of the global text prior. Without this, the exponential decaying problem would prevent windows that are distant in a naive flattened sequence from effectively interacting with the textual information.
> >
> > Thank you for your valuable suggestion regarding normalization before the SSM scans. We conducted an experiment where we added Layer Normalization (LN) layer immediately before the SSM scan in our WMF module, exactly as you suggested. As a result shown below, we observed a slight performance degradation with this addition compared to our original model without the LN layer. This suggests that our WMF module is inherently capable of performing effective cross-modal alignment without requiring additional normalization. The additional, external LN may have been redundant or slightly disruptive to this optimized flow.
> >
> > | Layer normalization before SSM | RefCOCO Val | RefCOCO TestA | RefCOCO TestB |
> > | :--- | :---: | :---: | :---: |
> > | O | 77.1 | 78.9 | 74.8 |
> > | **X (Default Setting)** | **77.3** | **79.2** | **74.8** |

---

> > ### Comment · Reviewer_WGQk · 2025-11-22
> >
> > Thanks for the comprehensive response, it has fully addressed my concerns. I will raise my score.

---

### Official Review · Reviewer_N61Y · 2025-10-31

**Soundness:** 2
**Presentation:** 2
**Contribution:** 2
**Rating:** 4
**Confidence:** 5

**Summary:**

The paper presents a parameter-efficient framework that integrates a window-based intermediate fusion neck (HMF) and lightweight adapters (MTA, MSA, and emphasis parameters) to enhance vision–language alignment for referring image segmentation.

**Strengths:**

- The paper introduces a Hierarchical Mamba Fusion (HMF) block, which performs intermediate vision–language fusion by aggregating multi-scale features and applying a window-based Mamba module (WMF).
- A parameter-efficient tuning (PET) strategy is presented, consisting of a Mamba Text Adapter (MTA) for modeling textual priors, a Multi-Scale Aligner (MSA) with RFMixer and cross-attention for visual–text alignment, and learnable emphasis parameters for adaptive layer weighting.
- The overall framework, WIMFRIS, integrates these components and is experimentally compared against existing PET-based and full fine-tuning methods on multiple RIS benchmarks.

**Weaknesses:**

* Lack of Novelty

The paper shows limited novelty. The **PET part** closely follows DETRIS, essentially extending its parameter-efficient tuning framework with minor Mamba-based modifications. The **neck design** heavily overlaps with the fusion architecture in fixation phase in SaFiRe, both adopting window-based Mamba fusion for intermediate vision-language alignment. Overall, the work mainly integrates these existing ideas rather than introducing a substantively new contribution.


* Incomplete Manuscript

The paper appears **incomplete**. Section 3.2 is unfinished, and the crucial description of the **task decoder** is missing. This omission disrupts the continuity between Sections 2.3 and 2.4. The authors should carefully verify whether the submitted version is the complete manuscript.


* Unfair and Limited Comparison

For Table 1


1. **Unfair Comparison :**
To ensure fairness, (1) the parameters of PET-based methods should be adjusted to achieve **comparable model sizes**, and (2) the **backbones of all compared methods** should be unified.

2. **Limited Comparison with State-of-the-Arts:**
More PET-based approaches should be included, as previous works (e.g., ETRIS, DETRIS, RISCLIP) have done, especially those involving **backbone-side modality fusion** in RIS, such as **PWAM in LAVT**, **SDF in VLT**, and **CFE in RISCLIP**, as well as classical parameter-efficient tuning methods like **LoRA** and **Adapter**.

3. **Marginal Improvement of the WMF Neck:**
Compared with **DETRIS**, the improvements achieved by the proposed **WMF Neck** are quite marginal.

4. **Insufficient Comparison :**
A more comprehensive comparison is needed to substantiate the claimed advantages of the proposed neck method, including detailed analyses of **parameter counts**, **computational cost (GFLOPs)**, and **inference speed**, particularly in comparisons with **ETRIS/DETRIS necks**.

For Table 2

1. **Inconsistent Metrics:**
   Table 2 mixes **mIoU** and **oIoU** without clarification. While RISCLIP, DETRIS, and WIMFRIS use **mIoU**, most other methods report **oIoU**. In particular, for works like **CGFormer** and **Polyformer**, which provide both metrics, the authors still report their **oIoU** values. Since **mIoU** is generally higher than **oIoU** on the RefCOCO family datasets, this inconsistency makes the performance comparison **unreliable**.
2. **RISCLIP Issue:**
   According to the authors’ own definition (line 44, “…keeping the vast majority of the backbone parameters frozen”), RISCLIP also freezes its CLIP backbone and should be considered a parameter-efficient tuning method. Moreover, the results of **RISCLIP-L** are missing, which appear **significantly higher** than those of the proposed “Ours-L” model (trained on RefCOCO+, mIoU: **RISCLIP-L** 74.38 / 78.77 / 66.84 vs. **Ours-L** 71.9 / 76.2 / 67.2).


*  Efficiency Analysis

Although this work emphasizes the **PET framework** and uses the **efficient Mamba architecture**, more detailed **efficiency analyses** should be provided—specifically **GFLOPs**, **inference speed**, and preferably **FPS**.


* Minor Issues

In **Table 3(a)**, the content does not match the caption: *4×4* is **not** the smallest window size.



***I would be happy to revise my score if the author addresses these points.***



---

**References:**

DETRIS: Densely Connected Parameter-Efficient Tuning for Referring Image Segmentation AAAI2025

SaFiRe: SaFiRe: Saccade-Fixation Reiteration with Mamba for Referring Image Segmentation NeurIPS 2025

LAVT: Language-Aware Vision Transformer for Referring Image Segmentation CVPR2022

VLT: Vision-Language Transformer and Query Generation for Referring Segmentation TPAMI2023

RISCLIP:Extending CLIP’s Image-Text Alignment to Referring Image Segmentation NAACL2024

LoRA: Low-Rank Adaptation of Large Language Models. ICLR2022

Parameter-Efficient Transfer Learning for NLP. ICML2019

CGFormer: Contrastive Grouping with Transformer for Referring Image Segmentation CVPR2023

PolyFormer: Referring Image Segmentation as Sequential Polygon Generation CVPR2023

**Questions:**

*  Could you clarify the **task decoder design**?

*  In Table 1, which IoU metric is used—**mIoU** or **oIoU**? ETRIS reports oIoU from the original paper, but DETRIS uses mIoU.

* In Table 2, please clarify metric issue and the RISCLIP issue mentioned in W1-B.

* What are the **inference speed** and **GFLOPs** of the proposed model?

---

> ### Author Response · Authors · 2025-11-21
> **Response to Reviewer N61Y**
>
> We sincerely appreciate you taking the time to review this article. The answers to each question are provided below.
>
> **Lack of novelty**
>
> Thank you for your insightful feedback. We appreciate the opportunity to clarify the novelty of our contributions, particularly regarding our PET framework and the WMF design.
>
> *On the novelty of the PET framework*
>
> Our framework and DETRIS [1] show fundamental differences in both motivation and architecture. First, regarding the text encoder adapter, DETRIS employs a 1D convolution-based structure to extract local text features, which inherently limits the receptive field to local context. In contrast, our Mamba Text Adapter (MTA) leverages an SSM-based Mamba block. We do not consider this a minor modification; it stems from a distinct motivation to model long-range dependencies among text tokens and generate an enhanced global textual prior. Second, regarding the vision encoder adapter, we distinguish our design from DETRIS through structural efficiency. DETRIS's Dense Aligner employs the D-MoC module, which relies on sequentially connected standard 2D convolutional layers. Conversely, our Multi-Scale Aligner (MSA), proposes the RFMixer module, which utilizes parallel branches of depth-wise strip convolutions. This parallelized design is structurally distinct from the sequential D-MoC and is specifically optimized to capture multi-scale context. Finally, as demonstrated in our ablation study (Table 3 (b)), each proposed component (MTA, MSA, and EP) contributes incrementally to the performance. We believe that these structural and functional distinctions validate that our PET framework is a novel contribution.
>
> *On the novelty of the WMF module compared to SaFiRe*
>
> We appreciate your reference to SaFiRe [2]. While the high-level concept of “window-based Mamba fusion” shares similarities, we would like to highlight distinct differences in architectural role, information flow, and core motivation. SaFiRe forms a single, long hybrid sequence by interleaving windowed sub-images with the full-text. In contrast, our WMF is motivated by the explicit goal of mitigating the exponential decaying problem in SSMs by creating multiple, short, parallel sequences by prepending a single global text token to each partitioned visual window, which is distinct from SaFiRe. Specifically, we rearrange the input feature map and text tokens into a tensor of shape $(B \cdot n_{win}, M+1, C)$. This formulation allows the SSM to process $(B \cdot n_{win})$ independent sequences simultaneously, each with a reduced sequence length of $(M+1)$, where $n_{win}$ denotes the number of windows and $M$ is the pixels per window. This ensures that every local visual window interacts directly and robustly with the global textual context without attenuation. In addition, SaFiRe’s Fixation module is applied sequentially and repeatedly across layers, whereas our WMF is a single module in HMF neck that aggregates multi-scale features once before the decoder. Furthermore, while SaFiRe focuses on refreshing textual information periodically, our primary motivation is to resolve the information decaying issue inherent in long-sequence scanning. Therefore, while both methods use window partitioning and Mamba, we believe that their information flow (parallel-short vs. interleaved-long), structural role (single neck vs. repeated backbone layer), and motivation (mitigating information decay vs. refreshing textual context) are different.
>
> **Incomplete manuscript and decoder design clarification**
>
> Thank you for pointing out this crucial point. We sincerely apologize for this significant omission and the confusion it has caused. To clarify, our task-specific decoder follows the architecture of CRIS [3], ETRIS [4], and DETRIS. Specifically, we leverage the vision-language decoder from DETRIS adapted with minor modifications, such as to dimensions and channel sizes, to fit our WMF structure. We have now completed the sentence to “We leverage the task decoder from DETRIS, replacing its cross-modal neck with our WMF.”

---

> > ### Author Response · Authors · 2025-11-21
> > **Response to Reviewer N61Y**
> >
> > **Unfair and limited comparison**
> > - Table 1
> > 1. Unfair comparison
> >
> > Thank you for highlighting this important issue. We agree that a fair comparison with a consistent number of PET parameters is crucial for demonstrating our model's effectiveness. To address this point, we performed an additional experiment to isolate the impact of our HMF neck. In this experiment, we fixed the parameter count to 1.4M (identical to ETRIS), using CLIP as text encoder and DINOv2 as vision encoder. The results below showed that incorporating our HMF neck provided a consistent performance improvement on the RefCOCO val split. This experiment validates that the performance gain is directly attributable to the architectural design of our HMF neck, rather than an increased number of parameters, since the PET parameter count and backbones were held constant. We believe this confirms the effectiveness of our proposed module.
> >
> > | Text enc. | Vision enc. | PET method | # Params (M) | Neck | RefCOCO Val | RefCOCO TestA | RefCOCO TestB |
> > | :--- | :--- | :--- | :---: | :---: | :---: | :---: | :---: |
> > | CLIP-ViT-B/16 [5] | DINOv2-B/14 [6] | ETRIS [4] | 1.4 | X | 72.2 | 73.9 | 70.8 |
> > | | | | | ETRIS | 74.5 | 76.5 | 72.9 |
> > | | | | | **Ours** | **75.7** | **77.5** | **73.3** |
> > | | | DETRIS [1] | 1.4 | X | 74.3 | 75.8 | 70.8 |
> > | | | | | DETRIS | 75.8 | 77.7 | 72.9 |
> > | | | | | **Ours** | **76.4** | **78.3** | **73.6** |
> > | | | **Ours** | 1.4 | **Ours** | **77.2** | **78.9** | **74.3** |
> >
> > 2. Limited comparison with State-of-the-Arts
> >
> > Thank you for your insightful comment. We conducted additional comparative experiments between our proposed framework and other PET frameworks. To ensure a fair comparison, we fixed the number of trainable PET parameters to 1.4M (identical to ETRIS) and used identical encoders, as well as the neck architecture. The results below showed that our proposed method yields optimal performance. This validates that the combination of our proposed PET framework and the neck structure is the most effective configuration. Please note that the CFE module of RISCLIP [11] was excluded as the lack of publicly available source code prevented accurate and fair comparison.
> >
> > | Text enc. | Vision enc. | PET method | # Params (M) | Neck | RefCOCO Val | RefCOCO TestA | RefCOCO TestB |
> > | :--- | :--- | :--- | :---: | :---: | :---: | :---: | :---: |
> > | CLIP-ViT-B/16 [5] | DINOv2-B/14 [6] | Adapter [7] | 1.4 | Ours | 73.5 | 75.4 | 70.0 |
> > | | | LoRA [8] | | | 74.5 | 76.7 | 71.1 |
> > | | | SDF [9] | | | 75.2 | 76.9 | 73.4 |
> > | | | PWAM [10] | | | 76.0 | 77.8 | 72.8 |
> > | | | ETRIS [4] | | | 75.7 | 77.5 | 73.3 |
> > | | | DETRIS [1] | | | 76.4 | 78.3 | 73.6 |
> > | | | **Ours** | | | **77.2** | **78.9** | **74.3** |
> >
> > 3. Marginal improvement of the WMF neck
> >
> > Thank you for this important observation. While the quantitative improvement over DETRIS appear modest in isolation, we emphasize that the core novelty of our WMF neck lies in its structural potential and synergistic role within the overall WIMFRIS architecture. From a structural perspective, our goal was to validate Mamba (SSM) as an effective alternative to attention-based mechanisms. Our results demonstrate that this novel SSM-based design can effectively replace existing necks without compromising performance, paving the way for more scalable RIS architectures. Furthermore, regarding synergistic aspects, our WMF neck is optimized to our PET framework. As the full WIMFRIS model achieves superior performance, our WMF neck plays a pivotal role in maximizing the utility of tine-tuned features. This improvement validates that WMF neck contributes to the overall robustness and effectiveness of our WIMFRIS.
> >
> >
> > - Table 2
> > 1. Inconsistent metrics
> >
> > Thank you for highlighting this important point. We found that the previous performance figures had been incorrectly recorded with a mix of oIoU and mIoU values. We have now revised the manuscript to reflect all performance metrics as mIoU.
> >
> > 2. RISCLIP issue
> >
> > Thank you for pointing that out. We have included the performance of RISCLIP-L [11] as requested. Regarding this comparison, we would like to offer clarification concerning the classification of RISCLIP as a parameter-efficient fine-tuning (PET) method. While freezing the backbone parameters is a primary characteristic of PET, the definitive classification also depends on the quantitative ratio of trainable parameters to the total model size. Although RISCLIP freezes the CLIP backbone, the paper does not disclose the specific parameter counts for its newly introduced CFE and SKE modules. Furthermore, as the source code is not currently available, it is different to verify the precise parameter overhead of these cross-attention-based components. Without quantitative evidence confirming that the added parameters remain within a highly efficient range, we believe it is premature to categorically classify RISCLIP as a parameter-efficient method in the same strict sense as established PET frameworks.

---

> ### Author Response · Authors · 2025-11-21
> **Response to Reviewer N61Y**
>
> **Efficiency analysis**
>
> Thank you for your valuable feedback regarding detailed efficiency analysis. We have measured and compared the GFLOPs, FPS, total inference time, and total number of trainable parameters for ETRIS, DETRIS, and our WIMFRIS on a single 24G RTX4090 GPU. (Please note that ETRIS uses CLIP as its vision encoder, while both DETRIS and our WIMFRIS use DINOv2.) The results of this analysis below show that when compared to DETRIS, although our WIMFRIS has a higher number of trainable parameters, it achieves significant performance improvement while maintaining a comparable computational cost (GFLOPs), inference speed and FPS. We believe this demonstrates that our performance gains are not simply from scaling, but are a result of the efficient architectural design of our WMF module.
>
> | Method | GFLOPs | FPS | Total Inference Time (sec) | Total Trainable Params (M) | Performance (RefCOCO-val) |
> | :--- | :---: | :---: | :---: | :---: | :---: |
> | ETRIS [4] | 64.85 | 15.81 | 241 | 25.9 | 70.4 |
> | DETRIS [1] | 128.67 | 10.87 | 350 | 25.5 | 76.0 |
> | **Ours-B** | **128.56** | **10.63** | **358** | **27.9** | **77.3** |
>
>
>
> **Minor issues**
>
> Thank you for pointing out this error. We have corrected “The smallest window size” to “The second smallest window size.”
>
> References
>
> [1] HUANG, Jiaqi, et al. Densely connected parameter-efficient tuning for referring image segmentation. In: Proceedings of the AAAI Conference on Artificial Intelligence. 2025. p. 3653-3661.
>
> [2] MAO, Zhenjie, et al. SaFiRe: Saccade-Fixation Reiteration with Mamba for Referring Image Segmentation. arXiv preprint arXiv:2510.10160, 2025.
>
> [3] WANG, Zhaoqing, et al. Cris: Clip-driven referring image segmentation. In: Proceedings of the IEEE/CVF conference on computer vision and pattern recognition. 2022. p. 11686-11695.
>
> [4] XU, Zunnan, et al. Bridging vision and language encoders: Parameter-efficient tuning for referring image segmentation. In: Proceedings of the IEEE/CVF international conference on computer vision. 2023. p. 17503-17512.
>
> [5] RADFORD, Alec, et al. Learning transferable visual models from natural language supervision. In: International conference on machine learning. PmLR, 2021. p. 8748-8763.
>
> [6] OQUAB, Maxime, et al. Dinov2: Learning robust visual features without supervision. arXiv preprint arXiv:2304.07193, 2023.
>
> [7] HOULSBY, Neil, et al. Parameter-efficient transfer learning for NLP. In: International conference on machine learning. PMLR, 2019. p. 2790-2799.
>
> [8] HU, Edward J., et al. Lora: Low-rank adaptation of large language models. ICLR, 2022, 1.2: 3.
>
> [9] DING, Henghui, et al. VLT: Vision-language transformer and query generation for referring segmentation. IEEE Transactions on Pattern Analysis and Machine Intelligence, 2022, 45.6: 7900-7916.
>
> [10] YANG, Zhao, et al. Lavt: Language-aware vision transformer for referring image segmentation. In: Proceedings of the IEEE/CVF conference on computer vision and pattern recognition. 2022. p. 18155-18165.
>
> [11] KIM, Seoyeon, et al. Extending clip’s image-text alignment to referring image segmentation. In: Proceedings of the 2024 Conference of the North American Chapter of the Association for Computational Linguistics: Human Language Technologies (Volume 1: Long Papers). 2024. p. 4611-4628.

---

### Author Response · Authors · 2025-12-01

Dear Area Chair,

We sincerely appreciate your efforts in managing the review process under the unprecedented circumstances.


We would like to summarize the constructive feedback provided by the reviewers, which we believe serves as a valuable reference for your decision-making.


Please kindly consider that we have thoroughly addressed all concerns by reviewers, leading to two reviewers explicitly raising their scores.


**Strength**


**Reviewer WGQK (Score 6, raised to 8)** highlighted that WIMFRIS achieves state-of-the-art or highly competitive performance across all standard RIS benchmarks (RefCOCO/+/g). The reviewer also acknowledged that the windowed approach to tackle the exponential decay in SSMs is well justified both mathematically and empirically.


**Reviewer CZEK (Score 4, raised to 6)** noted that the paper proposes a novel PET approach that achieves strong segmentation performance while enhancing efficiency through the HMF block leveraging the Mamba architecture. The reviewer positively evaluated the proposed modules, such as the SSM-based MTA and the MSA with RFMixer for handling multiple receptive fields, for their contribution to precise vision-language fusion.


**Reviewer UdVq (Score 6)** identified the introduction of a windowed SSM neck (WMF) to RIS as a key strength, noting it mitigates the exponential decay of vanilla Mamba. The reviewer also appreciated the novelty of the learnable emphasis parameters and the proven generic utility of the HMF module, which boosts performance when plugged into existing methods like ETRIS and DETRIS.


**Reviewer N61Y (Score 4)** positively mentioned the structural design of the HMF block, which aggregates multi-scale features and applies a window-based Mamba module. The reviewer also acknowledged the effective integration of the PET strategy, consisting of MTA (text adapter) and MSA (vision-text alignment), within the overall framework.


**Key Concerns Addressed**


*Fair comparison and scalability validation*


Concern: Questions were raised regarding whether the performance gains proceeded from the architectural design or simply from having more trainable parameters/different backbones compared to baselines (e.g., DETRIS, ETRIS).


Our response: We conducted an additional experiment fixing the number of trainable parameters to 1.4M (identical to ETRIS) and using consistent text/vision encoders. Even with the same parameter budget and backbones, WIMFRIS consistently outperformed existing methods on RefCOCO validation splits. This confirms that the improvements are directly attributable to the efficiency of our HMF neck and WMF module, not simply parameter scaling.


*Efficiency analysis (GFLOPs and FPS)*


Concern: A request for a detailed analysis of computational cost and inference speed to verify the "efficiency" claims.


Our response: We provided a comparison of GFLOPs, FPS, and inference time on a single RTX4090 GPU. WIMFRIS achieved significantly higher performance while maintaining comparable inference speed (FPS) and computational cost (GFLOPs) to DETRIS. This demonstrates the practical efficiency of our parallel window-based SSM design.


*Clarification of novelty*


Concern: Discussion was required to distinguish our work from recent concurrent works like SaFiRe and DETRIS.


Our response: We clarified that our Window Mamba Fusion (WMF) is structurally distinct. Unlike SaFiRe which interleaves sub-images, our WMF employs a parallel window partitioning strategy specifically motivated to solve the information decay problem in SSMs. Furthermore, our PET modules (MTA, MSA) utilize SSMs and parallel RFMixers for long-range and multi-scale modeling, offering a distinct structural advantage over the adapters in DETRIS.


*Robustness on Unseen Classes and Ablations*


Concern: Suggestions were made to evaluate generalization on unseen classes and to justify hyperparameter choices (e.g., window size, adapter placement).


Our response: We evaluated WIMFRIS on the "Unseen" split of the RefCOCO dataset. The results showed superior generalization compared to state-of-the-art methods. In addition, we added extensive ablation studies verifying that our default window size ($4 \times 4$) and adapter placement strategy offer the optimal trade-off between mitigating SSM decay and preserving local visual context.


*Manuscript Improvements*


We have revised the manuscript to standardize all metrics to mIoU for consistency, clarified the task decoder architecture , and refined the definitions of all abbreviations.


We believe these revisions solidly validate WIMFRIS as a framework that offers a novel, efficient, and robust solution for Referring Image Segmentation.

Sincerely, the authors

---

### Meta-Review · Area_Chair_aK7T · 2026-01-04

**Summary:**

WIMFRIS proposes a windowed Mamba-based fusion neck (HMF/WMF) plus a PET strategy for RIS and reports strong gains across RefCOCO/+/g. The rebuttal directly addresses major review blockers (incomplete decoder description, metric inconsistencies, fairness of parameter budgets/backbones, and efficiency/FLOPs/FPS), and two reviewers explicitly increased scores. Considering the technical contribution, the AC leans Accept. A suggestion is to additionally report results on GRES (gRefCOCO) to further validate robustness and generalization.

**Reviewer Concerns:**

Addressed: missing decoder/incomplete section fixed; metric inconsistency (mIoU vs oIoU) corrected; fair comparisons with matched PET params (1.4M) and added PET baselines (LoRA/Adapter/PWAM/SDF, etc.); added GFLOPs/FPS/inference time; added unseen-split generalization; added ablations for adapter placement, window size, and receptive-field configs; acknowledged and added failure-case discussion.

Still outstanding (minor): some novelty skepticism vs concurrent SaFiRe/DETRIS may persist (even if distinctions are clarified); practical limitations remain (needs fine-tuning labels; window partition can fragment large objects / miss cross-window context).

**Reviewer Scores:**

- WGQk: 6 → 7/8 (explicitly raised).
- CZEK: 4 → 6 (explicitly raised).
- UdVq: 6 → 6 (explicitly keeps score).
- N61Y: likely 4 → 5/6 (most listed blockers were addressed; novelty may keep it from going higher).

---

### Decision · Program_Chairs · 2026-01-26

Accept (Poster)